# Confinement discerns swarmers from planktonic bacteria

**Weijie Chen[1,2], Neha Mani[1], Hamid Karani[1], Hao Li[2], Sridhar Mani[2]\*, Jay X Tang[1]\***

[1]Department of Physics, Brown University, Providence, United States; [2]Department of Medicine, Albert Einstein College of Medicine, Bronx, United States

**Abstract** Powered by flagella, many bacterial species exhibit collective motion on a solid surface commonly known as swarming. As a natural example of active matter, swarming is also an essential biological phenotype associated with virulence, chemotaxis, and host pathogenesis. Physical changes like cell elongation and hyper-flagellation have been shown to accompany the swarming phenotype. Less studied, however, are the contrasts of collective motion between the swarming cells and their counterpart planktonic cells of comparable cell density. Here, we show that confining bacterial movement in circular microwells allows distinguishing bacterial swarming from collective swimming. On a soft agar plate, a novel bacterial strain *Enterobacter* sp. SM3 in swarming and planktonic states exhibited different motion patterns when confined to circular microwells of a specific range of sizes. When the confinement diameter was between 40 μm and 90 μm, swarming SM3 formed a single-swirl motion pattern in the microwells whereas planktonic SM3 formed multiple swirls. Similar differential behavior is observed across several other species of gram-negative bacteria. We also observed 'rafting behavior' of swarming bacteria upon dilution. We hypothesize that the rafting behavior might account for the motion pattern difference. We were able to predict these experimental features via numerical simulations where swarming cells are modeled with stronger cell–cell alignment interaction. Our experimental design using PDMS microchip disk arrays enabled us to observe bacterial swarming on murine intestinal surface, suggesting a new method for characterizing bacterial swarming under complex environments, such as in polymicrobial niches, and for in vivo swarming exploration.

**\*For correspondence:**
sridhar.mani@einsteinmed.org
(SM);
jay_tang@brown.edu (JXT)

**Competing interest:** See
page 17

**Reviewing editor:** Raymond E
Goldstein, University of
Cambridge, United Kingdom

## Introduction

Motility is an essential characteristic of bacteria. Although energy-consuming, it provides high returns, enabling cells to uptake nutrients efficiently and escape from noxious environments (*Webre et al., 2003*). In a host environment, bacterial motility is an essential phenotype that intimately relates to virulence through complex regulatory networks (*Josenhans and Suerbaum, 2002*). Swimming and swarming are two common motility phenotypes mediated by flagella. Whereas the planktonic phenotype defines individual bacteria's motility, a collective movement powered by rotating flagella on a partially solidified surface defines swarming (*Kearns, 2010*; *Partridge and Harshey, 2013*). In swarming, bacteria utilize their flagella to navigate through a thin layer of medium and uptake necessary molecules to maintain homeostasis and overall survival (*Darnton et al., 2010*). Morphological changes such as cell elongation and hyperflagellation may occur during swarming for some bacterial strains (e.g., *Proteus mirabilis*) (*Morgenstein et al., 2010*), but not others (e.g., *Photorhabdus temperata)* (*Michaels and Tisa, 2011*). Although concentrated swimming bacteria are often called 'a swarm of bacteria' without adhering to precise identification of swarming motility, most microbiologists believe that swarming and swimming are fundamentally different motility types. For instance, studies show that compared with swimming cells, the requirement for flagella torque is higher for swarming *B. subtilis* (*Hall et al., 2018*); swarming *E. coli* remodel their chemotaxis pathway (*Partridge et al., 2019*); and in swarming *P. aeruginosa*, both the production of

virulence factors and antibiotic resistance increase (*Overhage et al., 2008*). A recent study has demonstrated a medically relevant distinction between swarming and swimming: a particular strain of swarming *Enterobacter* protects against mice intestinal inflammation while its swarm-deficient counterpart does not (*De et al., 2021*). The evidence to date that shows swarming to be different from swimming comes mostly from biological data (*Kearns, 2010*). However, reliable characterization and quantitation of these differences are lacking. In this report, using *Enterobacter* sp. SM3, which is a novel strain that possesses both swimming and swarming motilities, we show distinct biophysical characteristics between these two types of motility under confined, circular geometry in a particular confinement size range.

Studies have shown that geometric constraints have a profound influence on patterns of microswimmers' collective motion. For example, these constraints may create mesoscopically or macroscopically coherent structures such as swirls and jets (*Theillard et al., 2017*; *Wioland et al., 2016a*; *Wioland et al., 2016b*). Circular confinement, in particular, could stabilize a suspension of motile bacteria into a spiral vortex (*Beppu et al., 2017*; *Lushi et al., 2014*; *Nishiguchi et al., 2018*; *Wioland et al., 2013*). Here, we compare the behaviors of bacteria in swarming and planktonic states under quasi-two-dimensional (quasi-2D) circular confinement. This new technical approach may be taken to detect bacterial swarmers from a given clinical sample. Such a new method of detection might lead to diagnostic applications since there are established associations between bacterial swarming and virulence pathology (*Lane et al., 2007*; *Overhage et al., 2008*).

## Results

### Swarming *Enterobacter* sp. SM3 forms large single swirls

A novel bacterial strain *Enterobacter* sp. SM3 (NCBI BioProject PRJNA558971), isolated in 2014 from mice with colitis induced with dextran sulfate sodium (DSS), has been previously studied for motility (*Araujo et al., 2019*) and host phenotype (*De et al., 2021*). SM3 expands rapidly on 0.5% agar with the collective motion of multilayers of cells at the edge. We mounted a PDMS chip containing circular microwells on agar surface. This technique allows us to observe bacterial motility for more than 3 hr (details with illustration in Materials and methods). Under confinement in circular wells in the diameter range of 31–90 µm, swarming SM3 cells form single swirls. In contrast, SM3 planktonic cells concentrated from the liquid medium form mesoscale vortices (multiple swirls) in the same size range, except for the smallest well diameter of 31 µm. A clear difference is shown at the well diameter of 74 µm (*Figure 1A–D*, *Videos 1* and *2*). This striking difference persists in several well depths, except that the concentrated cells yield small but non-zero vortex order parameters (VOPs, defined as illustrated in *Figure 1E*) in deeper wells, instead of nearly zero VOPs in shallow wells (*Figure 1F*).

The confinement diameter has a strong influence on the motion pattern in the wells. In smaller wells such as ones of 31 µm in diameter, even concentrated planktonic SM3 forms a single vortex (*Figure 2A*), whereas in larger wells, such as ones of 112 µm in diameter, swarming SM3 also breaks into mesoscale vortices (*Figure 2B*). The phase diagram shows a single swirl in small confinement for both phenotypes of SM3. As the confinement size increases, the VOP of planktonic SM3 drops as the motion pattern breaks into multiple vortices. The drop of VOP and occurrence of multiple vortices occur to swarming SM3 at much larger sizes (*Figure 2C*). To further compare the dynamics of the confined swarming and planktonic SM3, the spatial correlation of the velocity field was calculated for d = 90 µm (where the motion patterns differ for swarming and planktonic SM3) and for d = 500 µm (where both motilities show mesoscale vortices) (see Materials and methods). We computed the correlation function for an inscribed square within a well (e.g., a square of 60 µm × 60 µm for d = 90 µm confinement), which shows the extent to which the velocity at an arbitrary location correlated with the velocity at a distance of ∆r away from that location. In 90 µm wells, swarming SM3 velocity correlates positively or negatively throughout the whole well (negative values have resulted from the opposite sides of a single swirl). In contrast, the swimming velocity of planktonic cells of comparable concentration does not correlate once ∆r > 25 µm (*Figure 2D*). However, in a large open space where both swarming and swimming SM3 break into small vortices, the correlation functions fall into similarly low values. The velocity correlation length as the curve first crosses $C_r(\Delta r)$

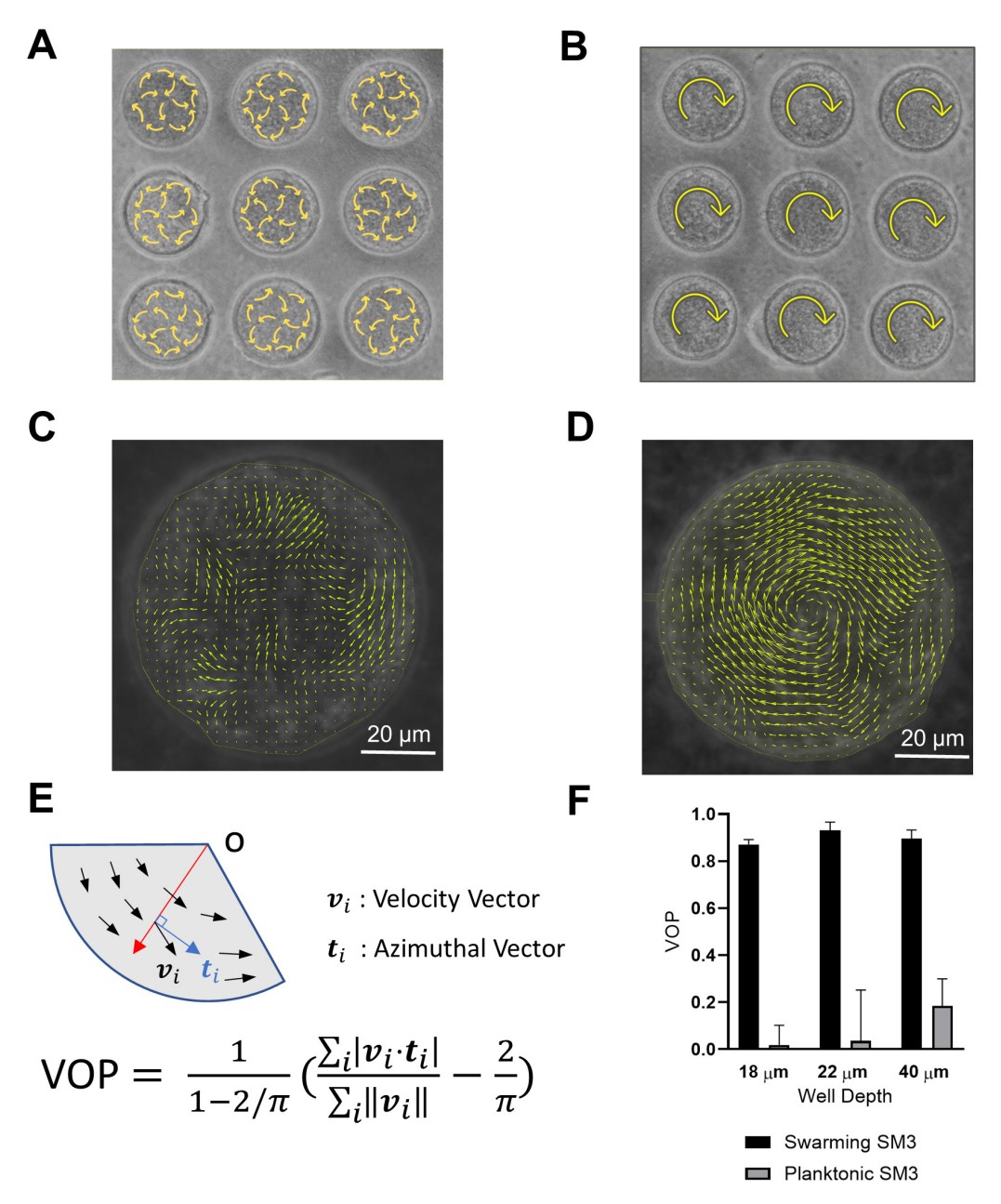

**Figure 1.** Swirls of *Enterobacter* sp SM3 under circular confinement. (**A and B**) Motion patterns of concentrated planktonic (**A**) and swarming (**B**) SM3 in the PDMS microwells of 74 µm in diameter. Circular arrows indicate the direction of bacterial collective motion. (**C and D**) Velocity field of concentrated planktonic (**C**) and swarming (**D**) SM3 in a single microwell. (**E**) Illustration of how vortex order parameter (VOP) is defined. |·| denotes the absolute value while ‖·‖ denotes the Euclidean norm. (**F**) VOP of swarming and swimming SM3 in 74 µm microwells of three different depths. The sample size n = 5 for each group and the values are represented as mean and standard deviation (± SD).

=0, which represents the size of the mesoscale vortices, is 23 µm and 28 µm for planktonic and swarming SM3, respectively (*Figure 2E*).

## The large single-swirl behavior is indicative of cell–cell interaction

We performed several experiments to explore parameters that might have caused the divergence of motion patterns between swarming and concentrated planktonic cells in confinement. First, we ruled out cell density difference as the reason for the difference in the confined motion patterns by

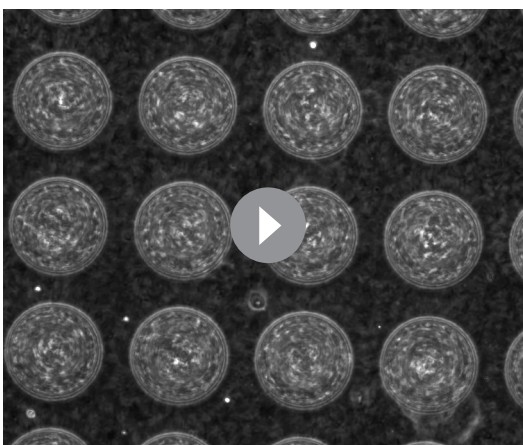

**Video 1.** Confined swarming SM3 showing a single-swirl motion pattern. Swarming SM3 cells were confined in 74 μm diameter PDMS wells. Video plays in real-time.

https://elifesciences.org/articles/64176#video1

concentrating planktonic cells to a comparable density of a naturally expanding swarm on agar (see Materials and methods) before mounting the PDMS chip. Second, we noticed that SM3 tends to get elongated when they swarm (*De et al., 2021*). We hypothesized that elongated bacteria may enhance the local alignment of the rod-shaped cells and increase the vortex size in meso-scale turbulence (*Doostmohammadi et al., 2016*; *Peruani et al., 2006*). Thus, we treated SM3 planktonic cells with cephalexin (CEP) which has been shown to elongate *E. coli* (*Hamby et al., 2018*). This treatment indeed caused the cell length of SM3 to reach that of swarming cells on average (*Figure 3A*). However, we found no significant change in swimming speed or VOP when confined in comparably high cell density following the centrifugation and CEP treatment of the planktonic SM3 (*Figure 3B and C*). Although CEP-treated planktonic SM3 has similar cell length, cell density, and cell speed as swarming SM3, we could not restore the single-swirl pattern in 74 μm confinement wells (*Figure 3C*). Third, noticing a surfactant rim on the swarming SM3 colony edge, we conjectured that surfactants secreted by swarming SM3 might help align the swarmers in confinement. As a prototypical surface wetting agent, surfactin was added in several concentrations to planktonic SM3 to test whether it could promote a single-swirl motion pattern. However, it did not establish a stable single-swirl pattern. Finally, we found that adding lyophilized swarming supernatant to swimming SM3 did not increase the VOP either (*Figure 3C*).

While unable to make the concentrated planktonic SM3 form a single swirl in the 74 μm well, we tackled the problem from another angle, by altering the conditions of swarming SM3 in order to break the single swirls. Initially, we tried to physically 'disrupt' the swarming colony by rubbing the swarming colony gently with a piece of PDMS offcut roughly 0.5 cm × 1 cm in size. This operation did not break the single-swirl pattern in the wells (*Figure 3D*). Then, 0.2% D-mannose was added to the swarming colony to de-cluster bacteria bundles due to cells' sticking to each other (*Hamby et al., 2018*). However, this treatment could not alter the single-swirl pattern either (*Figure 3D*).

Finally, we tried replacing the swarm fluid by diluting the swarming cells in Lysogenic Broth (LB) by 100-fold and then reconcentrated the cells in order to test if the single swirl pattern would still form. This was done by centrifuging at 1500 g for 10 min and then removing extra LB to recover the initial cell density. These 'rinsed' swarming SM3 cells were pipetted back on the agar plate. This process was expected to wash away some extracellular matrix polymers, including perhaps those weakly adhered on the bacterial surface but that would unbind upon dilution. As shown in *Figure 3B*, this fluid replacement treatment did not alter the cell motility significantly. However, after this treatment, we observed multiple swirls under the confinement that previously produced single swirls

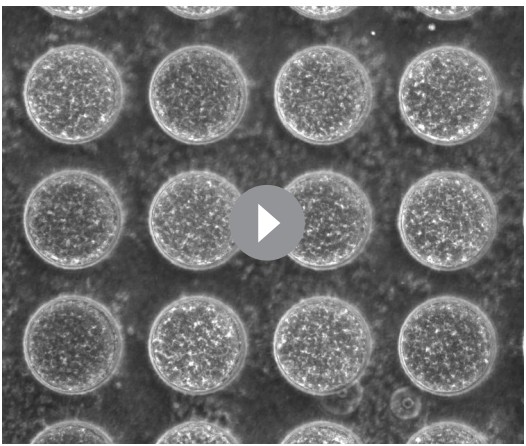

**Video 2.** Confined concentrated planktonic SM3 showing a turbulent motion pattern. Swimming, planktonic SM3 cells were confined in 74 μm diameter PDMS wells. Video plays in real-time.

https://elifesciences.org/articles/64176#video2

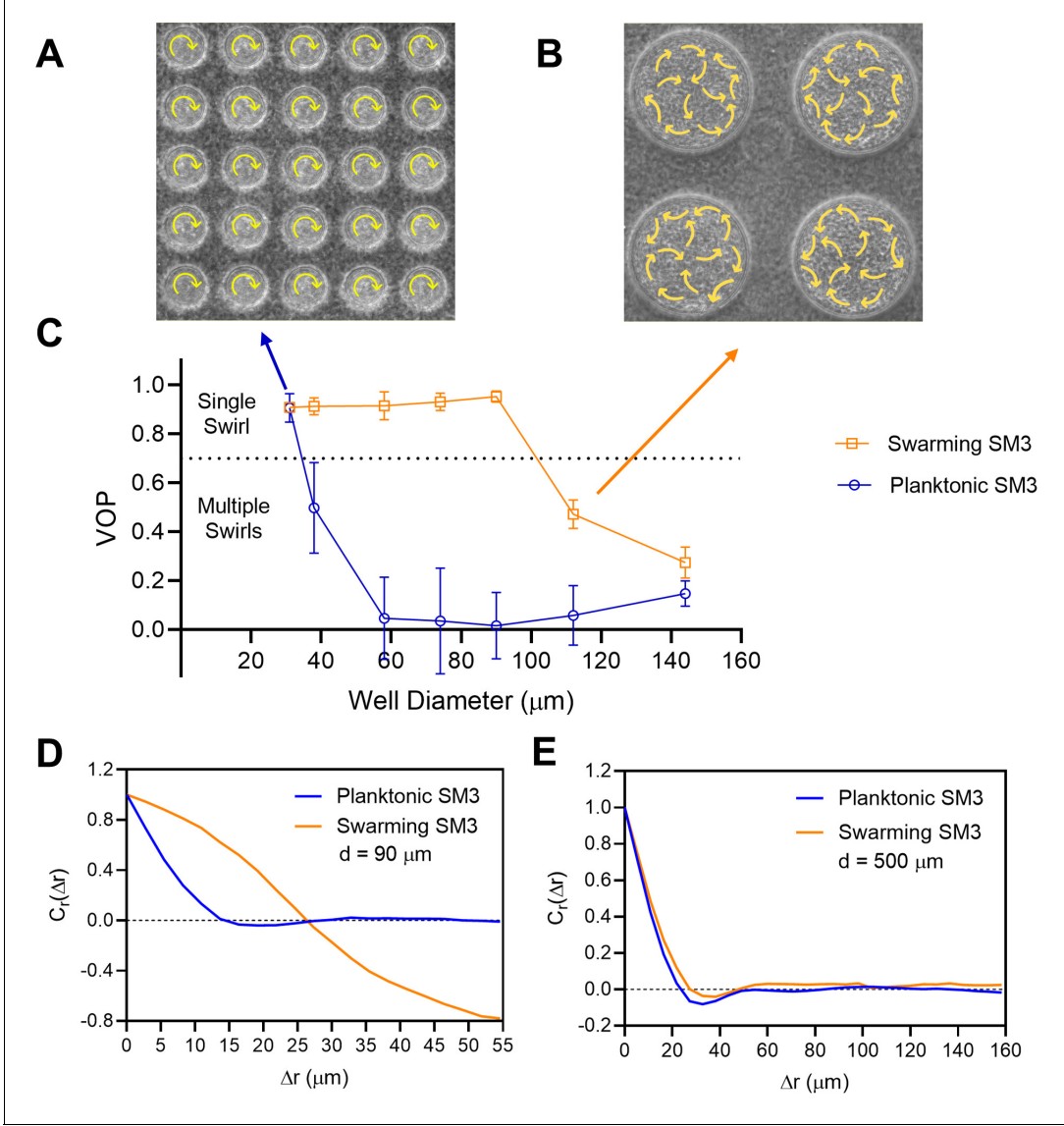

**Figure 2.** The effect of well diameter on confined *Enterobacter* sp SM3 motility patterns. (**A and B**) Motion pattern of concentrated planktonic SM3 confined in 31 μm (**A**) and swarming SM3 confined in 112 μm (**B**) diameter microwells. (**C**) VOP of swarming and concentrated planktonic SM3 as a function of well diameter. The error bars represent standard deviations (± SD), and the sample size is n = 5. (**D and E**) Spatial autocorrelations of the bacterial velocity field in the well diameters of 90 μm (**D**) and 500 μm (**E**). The depth of the wells was 22 μm for all well diameters shown in this figure.

(*Figure 3D*), suggesting that these 'rinsed' cells behave much like planktonic cells. These results suggest that the single swirl pattern depends on extracellular materials removable by matrix dilution. These extracellular materials act to affect matrix rheology and, perhaps more directly, cell–cell interaction, causing changes in the swirl pattern.

## Diluted swarming SM3 show unique dynamic clustering patterns

We suspected that specific interactions between the neighboring swarming cells were weakened or diminished upon dilution with the LB medium. A 50 μL water droplet was applied to the swarming and the concentrated planktonic SM3 colony edges to investigate intercellular interaction at a microscopic scale within the bacterial colony. In the diluted swarming colony, groups of cells formed bacterial rafts, a characteristic feature previously associated with gliding motility (*Be'er and Ariel, 2019*; *Kearns, 2010*). Those cells within a polar cluster are moving in the same direction in a cohesive pack at the same speed (*Video 3*). In contrast, upon dilution of the concentrated planktonic SM3, the cells

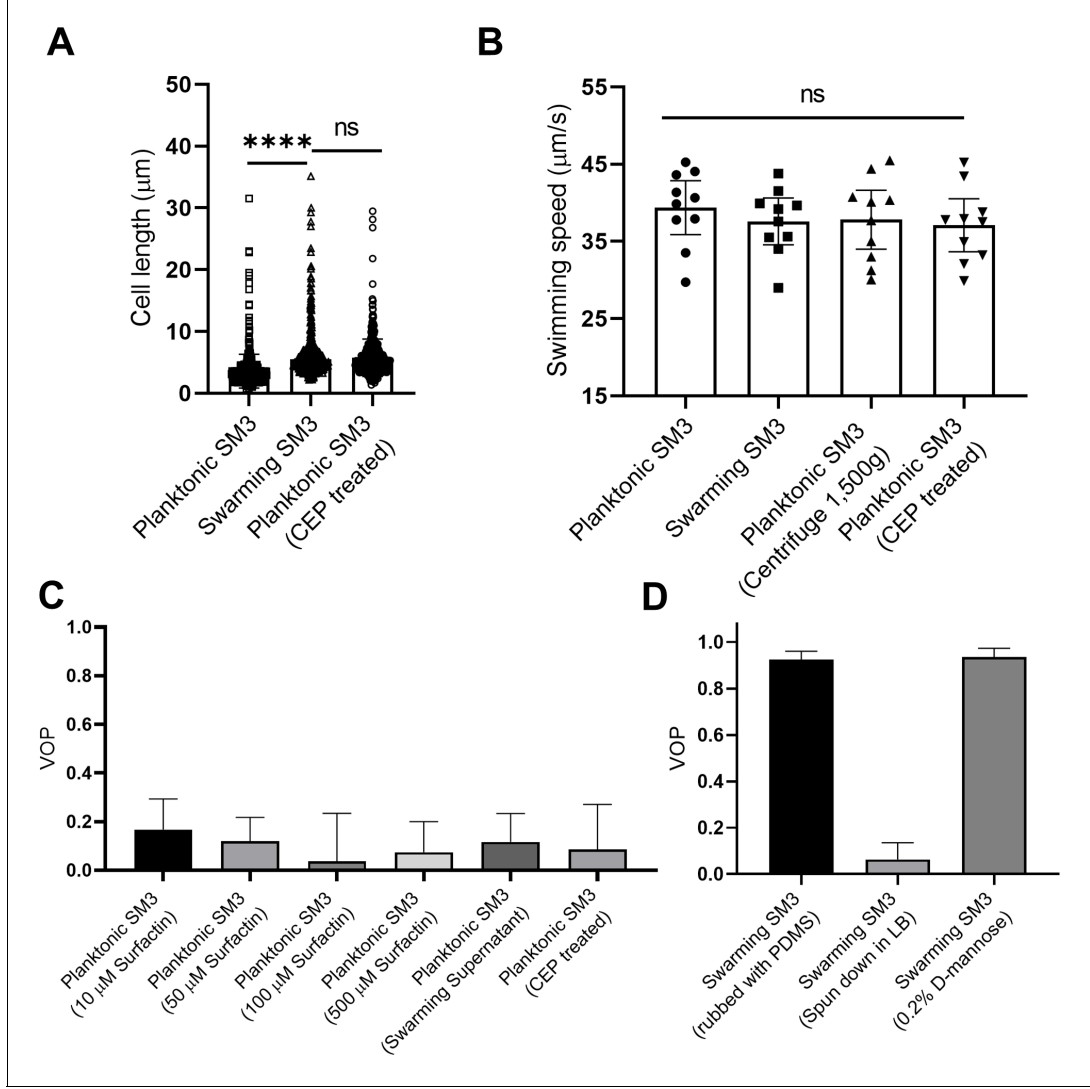

**Figure 3.** Factors that possibly influence the bacterial motion pattern in the well. (**A**) Bacterial cell length of planktonic, swarming, and cephalexin (CEP) treated planktonic SM3, n = 500 for each group. Data are represented as median and interquartile range. **** indicates p<0.0001. ns indicates not significant, with p=0.8755 (Kruskal–Wallis test). (**B**) Bacterial cell speed of swimming, swarming, centrifuged, and CEP-treated swimming SM3, n = 10 for each group. p=0.7375, as determined by one-way ANOVA followed by Tukey's post hoc test; ns, not significant. (**C**) VOP of swimming SM3 under 74 μm diameter confinement with different treatments, n = 5 for each group. (**D**) VOP of swarming SM3 under 74 μm diameter confinement with different treatments, n = 5 for each group. **B–D**, Data are shown as means with standard deviation (± SD) indicated. All statistical tests were performed using GraphPad Prism v8.4.3.

disperse uniformly, and their moving directions appear random (*Video 4*). Swarming SM3 cells tend to move together near the agar surface, while planktonic SM3 cells swim freely in the bulk fluid (*Figure 4A and B*). We used the MATLAB PIV toolkit to track the moving bacteria in the image sequences of diluted swarming and planktonic SM3 for comparison. We found that swarming SM3 formed clusters with more than 20 cells on average, while we did not see such clusters of planktonic SM3 cells (*Figure 4C and D*). The lingering clusters of cells in the swarming phase upon dilution point to a more substantial cell–cell interaction than between planktonic cells.

## Numerical simulation reveals cell–cell interaction to be the key factor for large swirls

To further verify that rafting in swarming is a crucially relevant factor to the motion pattern discrepancy, we performed computer simulations using a zonal model for pair-wise interactions. The

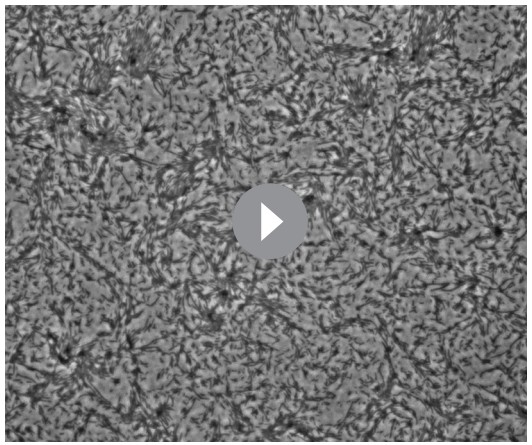

**Video 3.** Diluted swarming SM3 colony. The swarming SM3 colony edge was diluted by adding a 50 µL water droplet. Clusters of bacterial cells formed rafts. Video plays in real-time.

https://elifesciences.org/articles/64176#video3

interactions among the moving particles (short-range repulsion, velocity alignment, and anti-alignment) are considered, all as functions of the particle–particle distance (*Grossmann et al., 2014*; *Grossmann et al., 2015*). Based on the data shown in *Figure 3B*, the particles' swimming speed is fixed to 37 (µm/s), but the initial particle positions and initial moving directions are randomized. In the simulations, we interpret the rafting as due to a more substantial alignment among the swarmers, which is captured through a larger alignment interaction range (described in Materials and methods and in Appendix 2). This treatment results in longer correlation length of swarmer cells in the velocity auto-correlation plots shown in *Figure 2D and E*. This longer correlation length persists in all confinement sizes, even at the largest domain size of 500 µm, where the correlation length of swarmer cells is approximately 5 µm longer than planktonic cells, that is, ~28 µm vs. ~23 µm. This notable difference prompted us to differentiate swarmer and planktonic cells using two different alignment interac-

tion ranges, i.e., 20 µm and 15 µm for swarmer and planktonic cells, respectively (See *Appendix 2—table 1* for a complete list of simulation parameters). The number of particles in our 2D simulations is calculated from the average number of cells found in microwells of different sizes. It was assumed based on the bulk cell density estimate that cells form pseudo two-dimensional layers of 4–5 µm thickness in each 22 µm deep cylindrical microwell (See Appendix 2 for detailed explanation).

We simulated the motion of confined swarmers and planktonic cells in different sizes of circular confinement similar to those in the experiments. By solely assigning an enhanced alignment interaction range for swarming cells, the simulation results mirror the experimental results. Both swarmers and planktonic cells start with a single-swirl pattern at the smallest disk size of 31 µm; as the confine-

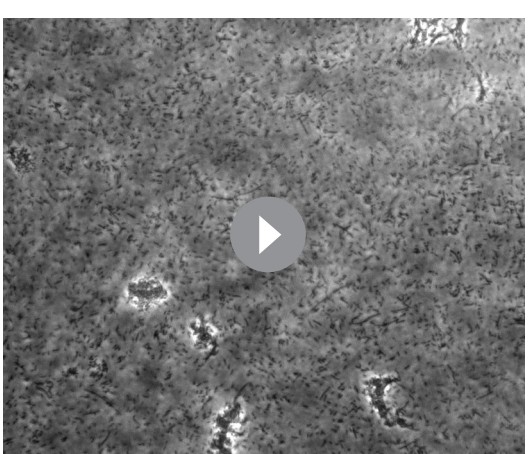

**Video 4.** Diluted swimming SM3 suspension. Concentrated sample of planktonic SM3 spread on agar was diluted by adding a 50 µL water droplet near the edge. Cells in the diluted region were observed to swim independently without clustering. Video plays in real-time.

https://elifesciences.org/articles/64176#video4

ment size increases to 38 µm, the planktonic cells break into a multi-swirl motion pattern, whereas the swarmers remain in single swirls until more than twice the disk diameter. At the confinement size of 90 µm, disks filled by swarmer cells transition into multi-swirls. At large enough size, multi-swirl patterns occur to samples of both cell types (*Figure 5A*, compared with *Figure 2C*; also see *Appendix 2—figure 2* and *Video 5*). The velocity auto-correlation of simulation data for confinement sizes of 90 µm and 500 µm (*Figure 5B and C*) shows quantitative agreements with corresponding experimental findings (*Figure 2D and E*). Consistent with experiments, velocity correlation length of multi-swirl patterns formed by swarmer cells is a few microns larger than those in planktonic cells. We then performed the 'dilution' simulation for both states, finding that swarming cells form dynamic polar clusters at the cell densities smaller than the concentrated case in the confined system. In contrast, the planktonic cells form a 'gas' phase without clustering at all comparable densities (*Figure 5B*, *Video 6*). This result supports the

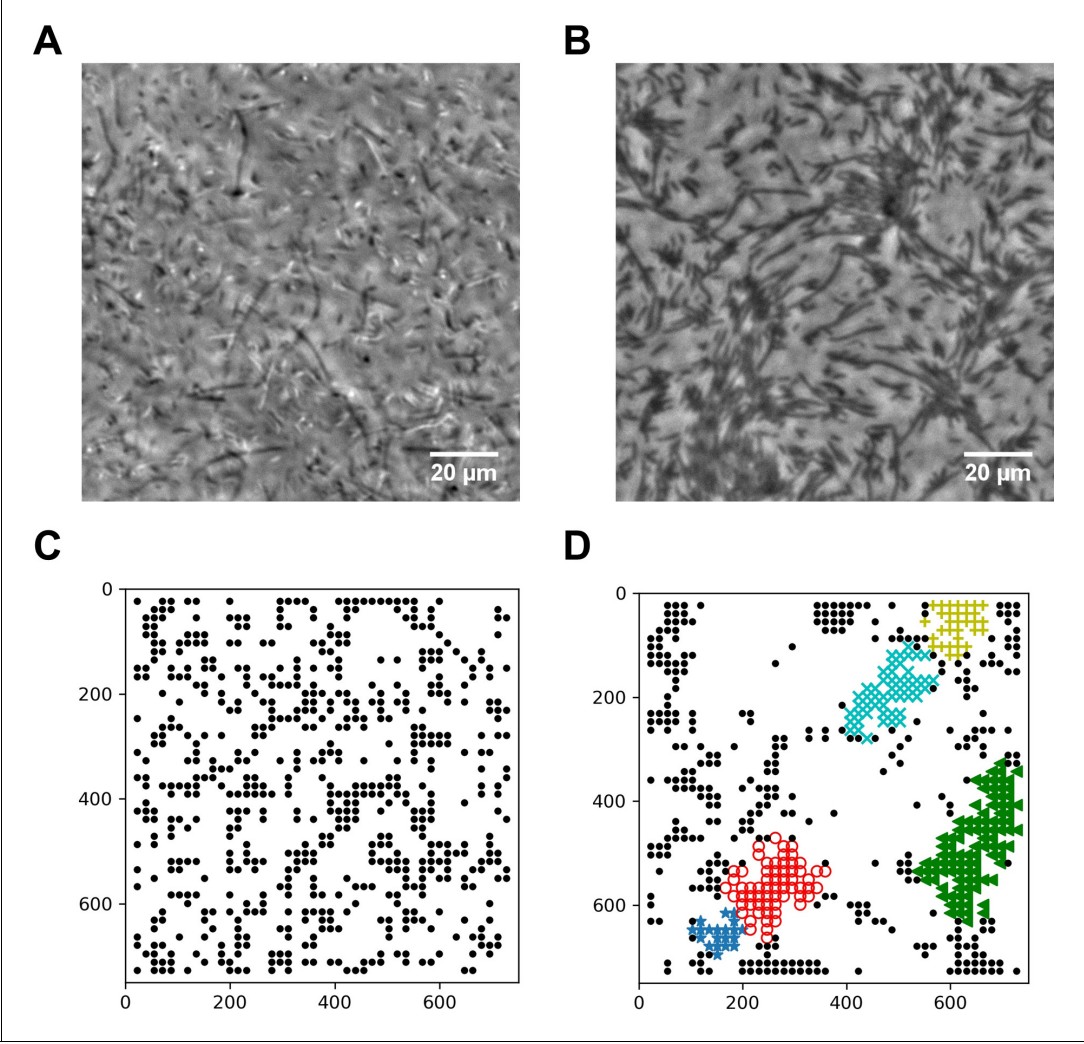

**Figure 4.** Spatial distribution of swarming and swimming SM3 cells. (**A and B**) Snapshots showing diluted swarming SM3 (**A**) and planktonic SM3 (**B**) on a soft agar surface, respectively. (**C and D**) DBSCAN clustering analysis of diluted swarming SM3 (**C**) and planktonic SM3 (**D**). Black dots represent moving bacterial cells and colored markers show cells in clusters, as determined by the program. The numbers on the axes correspond to pixels on the images.

experimental observation in *Figure 4A*. In short, the simulation captured the motion pattern features (*Figure 5D and E*; *Figure 5—figure supplement 1*) and predicted the experimental results in both confinement and dilution experiments.

## Identifying SM3 motility type on mice mucosal surface

The difference in confined motion patterns enables us to detect bacterial swarming on surfaces other than agar, including under physiological environments such as on mucosal surfaces. Unlike experiments on an agar surface, there are considerable technical challenges in dealing with uneven or more complex surfaces. The mouse intestinal tissue, for instance, is more than 1 mm thick and non-transparent. Since light cannot penetrate the tissue, observing bacteria directly on the inner surface through the tissue is not feasible. Staining or fluorescence labeling may alter the bacterial swarming motility (e.g., we found that SM3 becomes non-swarming once GFP labeled). If the bacterial cells are labeled biochemically, the fluorescence signal weakens when the cells reproduce. As an alternative strategy, using PDMS chips coated with fluorescent beads and then mounted on SM3 inoculated C57BL6 mouse intestine tissue, we detected swarming motility based on the 'collective' swirling motion of the beads (see Materials and methods, *Figure 6*, and *Videos 7* and *8*). This

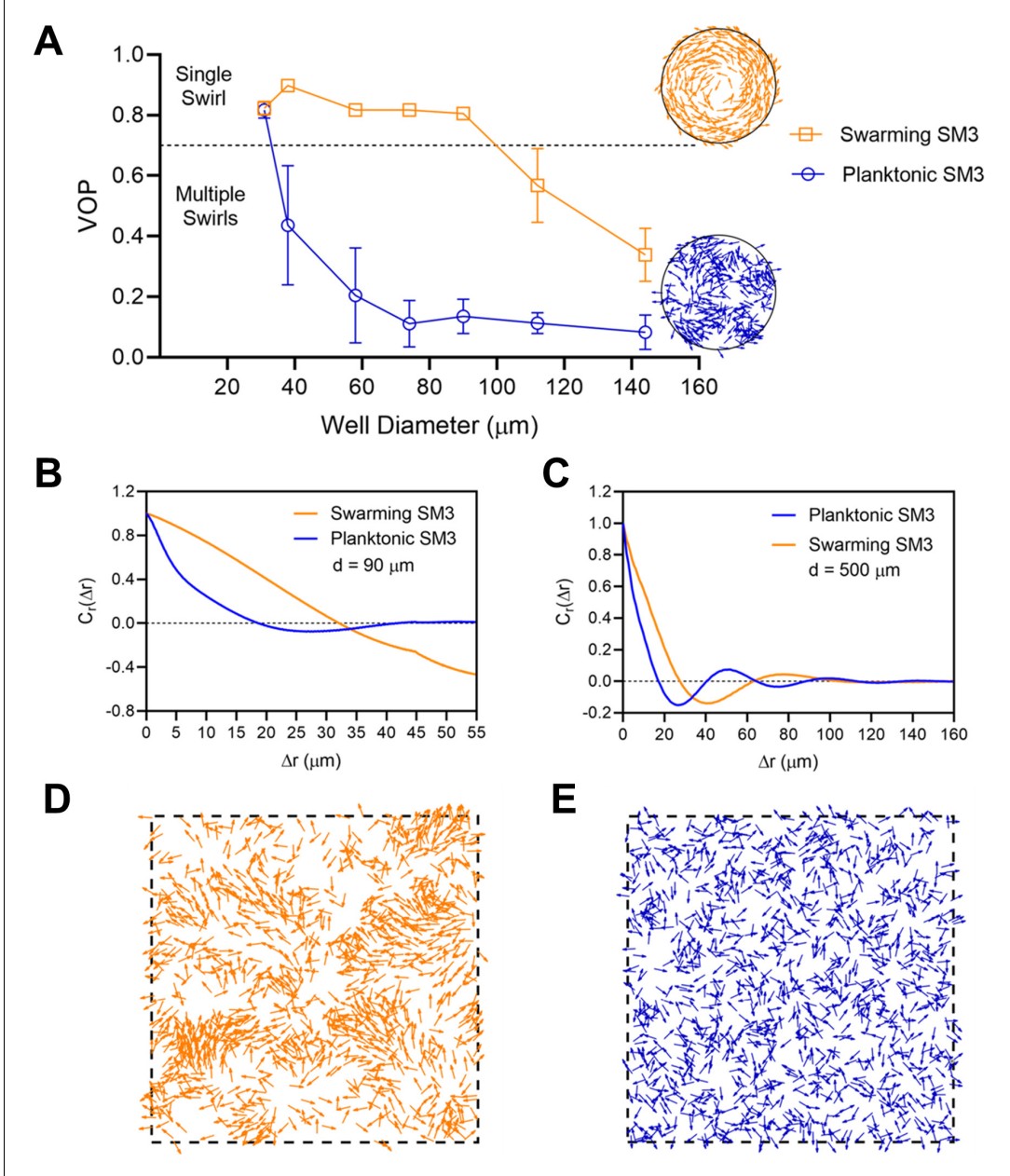

**Figure 5.** Numerical simulations of planktonic and swarming SM3 in confinement and open space. (A) VOP of swarming and concentrated planktonic SM3 as a function of well diameter. The error bars represent standard deviations (± SD), and the sample size is n = 5. The circles on the right show representative motion patterns of swarmers and concentrated planktonic cells in the confinement size of 90 μm. (B and C) Spatial autocorrelations of the bacterial velocity field in the well diameters of 90 μm (B) and 500 μm (C), respectively. (D and E) Diluted swarmer cells (D) and diluted planktonic cells (E) with the same cell density in a space of periodic boundary condition.

The online version of this article includes the following figure supplement(s) for figure 5:

**Figure supplement 1.** Velocity field of swarming SM3 in 500 μm diameter confinement.

experiment on the mouse intestine tissue confirms that bacterial swarming indeed occurs on a physiologically relevant surface, much like on the commonly observed agar surface.

## Discussion

We have shown the motion pattern differences between PDMS chip confined planktonic and swarming *Enterobacter* sp. SM3 in the size range of 40 µm ≤ d ≤ 90 µm. Compared with previous work, our experimental setup has the advantage of ensuring stable and sustainable patterns. First, PDMS material does not harm living bacteria cells and is permeable to oxygen (*Turner et al., 2010*), thus ensuring continued oxygen exposure required for swarming (*De et al., 2021*). Second, we mounted the microchip on a soft agar containing over 97% water by mass, which automatically fills the wells via permeability and capillary flow. Finally, the LB agar also provides the necessary nutrients to fuel the bacterial growth and movement in the wells. Therefore, bacterial cells confined in the microwells remain motile for

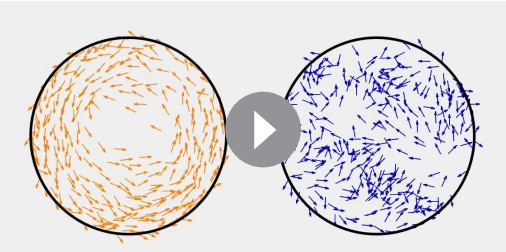

**Video 5.** Numerical simulations of circularly confined SM3. Swarming SM3 (left) and concentrated planktonic SM3 were simulated in the well size of 90 µm. The video shows a representative confined motion pattern. Arrows indicate the moving direction of the particles. Video plays in real-time.
https://elifesciences.org/articles/64176#video5

hours, much longer than in droplets surrounded by mineral oil (*Hamby et al., 2018*; *Wioland et al., 2013*) or in microfluidic chambers with glass surfaces (*Beppu et al., 2017*; *Wioland et al., 2016a*), where bacterial movement typically lasted no more than 10 min.

One interesting observation is that the rotation direction of the single swirls in our system is clockwise biased (85%). We interpret this bias of swirl direction as a consequence of flagellar handedness. When confined between the agar and PDMS surfaces, bacteria tend to swim closer to the porous agar surface (*DiLuzio et al., 2005*). For a bacterium swimming near the agar surface, there is a sideway component of drag force experienced by the rotating cell body. Similarly, there is a sideway component acting on the flagella bundle at the tail end of the cell in the opposite direction. These two sideway forces result in a net torque on the bacterium, causing it to make gradual right turns (*Araujo et al., 2019*; *DiLuzio et al., 2005*). In our case, the agar surface is the bottom surface, and the bacteria tend to follow a clockwise curved trajectory near the bottom (when viewed from the top). This effect may persist in collective cell motion for the single swirls to appear as clockwise more often than counterclockwise. The effect in the concentrated swirls is notably weak, in view of the exceptions occurring in nearly 15% cases. The bias in handedness must be abrogated in cases of multiple swirls, or when another dynamic effect becomes dominant (*Chen et al., 2017*).

Prior studies have proposed different models to explain the circularly confined motion of rod-shaped swimmers (*Hamby et al., 2018*; *Lushi et al., 2014*; *Tsang and Kanso, 2015*). In our case, we adopt the Zonal model (*Grossmann et al., 2015*) in order to explain the motion pattern difference observed for confined swarming and planktonic SM3. Noticing that swarming SM3 washed in LB lost the single-swirl pattern, we hypothesize that other than cell length or cell speed, the strong cell–cell interaction may be a key factor responsible for the persistence of single swirls in the wells. The mechanism of the rafting phenomenon of swarming cells has not been fully deciphered yet (*Kearns, 2010*). It might be due to cohesive interaction among neighboring cells and/or hydrodynamic effects among 2D-confined peritrichous flagellated bacteria (*Li et al., 2017*). The cell–cell interaction may be attributable to biochemical changes of cell envelope during swarming (e.g., more long sidechain lipopolysaccharides) or secretions (*Armitage et al., 1979*). Once these surrounding

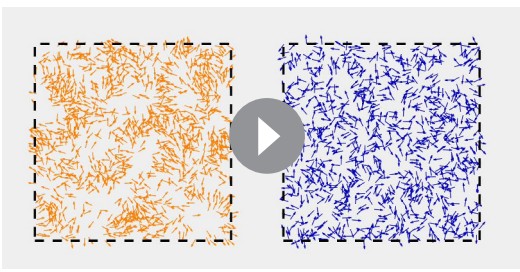

**Video 6.** Numerical simulations of SM3 cells in open space. Diluted swarming SM3 (left) and planktonic SM3 were simulated without confinement, but with a periodic boundary condition. In both cases, cell density is ρ = 0.025 µm$^{-2}$ (~10 cells in a 20 µm × 20 µm area) and the arrows indicate the moving directions of the particles. Video plays in real-time.
https://elifesciences.org/articles/64176#video6

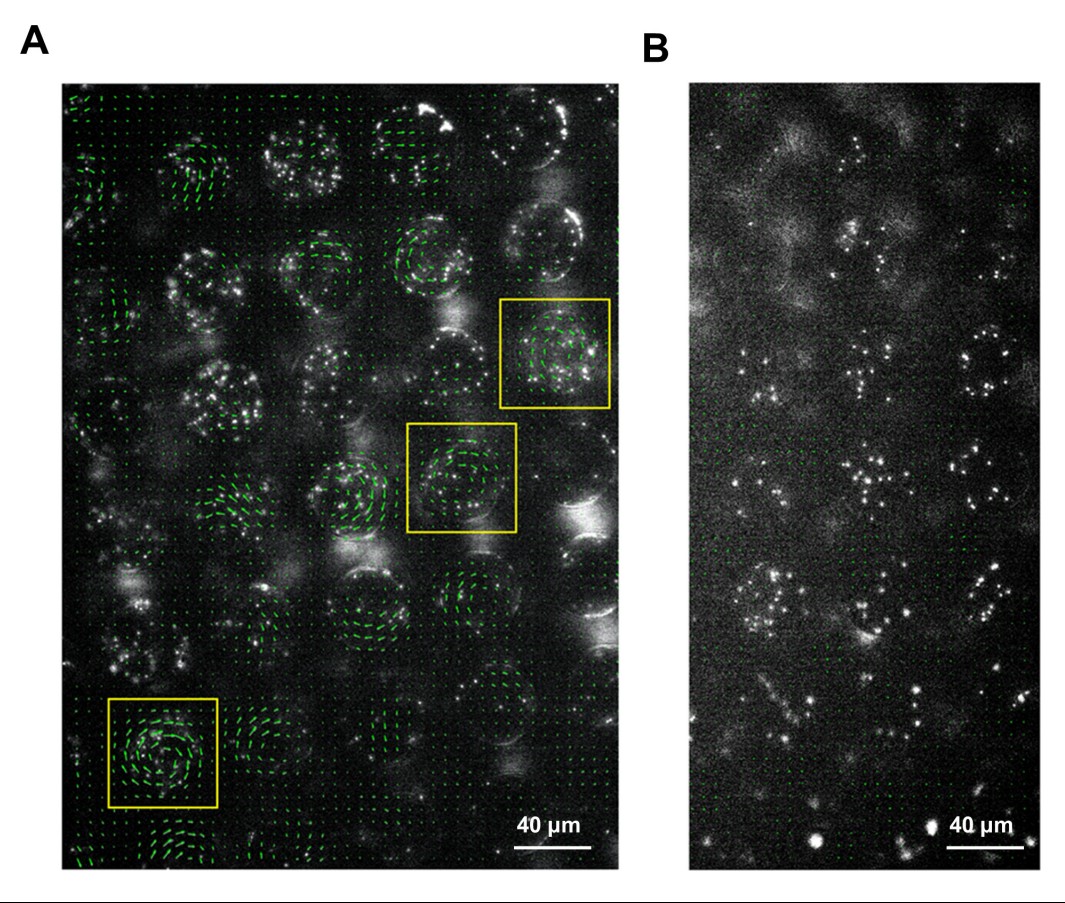

**Figure 6.** Motion of fluorescent beads in microwells mounted on murine tissues. PDMS chips were coated with 0.5 μm fluorescent beads and mounted on SM3 inoculated colitic (**A**) or non-colitic (**B**) mice intestinal tissue surfaces. The beads' motion was measured after 4.5 hr incubation. Average velocity field was calculated by tracing the beads' motion using PIV toolkit. (**A**) On colitic tissue, wells with VOP > 0.7 were found and marked with yellow squares. In these wells, the single swirl motion pattern of the beads was powered by the confined swarming bacteria. Since the tissue surface was not as smooth as on agar surface, the motion of the beads in some wells did not form a complete vortex, yet jets indicating partial vortices were discernable. (**B**) On a normal tissue lacking swarming bacteria, the average velocity of the beads in the wells due to random motion was close to zero, giving rise to uniformly small VOP values. We could infer that the confined SM3 in these wells were predominantly swimming rather than swarming.

matrix or polymers are washed away by ~100-fold dilution, the cohesive interactions may diminish, resulting in a loss of dynamic clusters upon 100× dilution. After being concentrated back to comparable cell density and subjected to the same confinement, these rinsed swarming bacteria formed multi-swirl motion patterns as opposed to the more coherent single swirls. However, it is also possible that local viscosity drops dramatically in the environment of the swarmers after being washed. Then, the momenta of nearby cells could not be passed to each other as efficiently as previously, resulting in the absence of dynamic clusters and the loss of single-swirl motion pattern. Indeed, a recent publication shows that augmenting the viscoelasticity of dense swimming *E. coli* alone (by adding purified genomic DNA) can turn bacterial turbulence into a giant, single vortex in circular confinement (*Liu et al., 2021*).

In multi-swirl patterns, we consistently observed that velocity correlation length of swarmer cells is a few microns longer than in planktonic cells in all sizes of confinement. We incorporate this observation in our simulation by assigning a larger alignment interaction range $r_a$ for swarmer cells. We confirm that enhanced alignment is a key factor that differentiates swarmers from planktonic cells. The simulation results show that enhanced alignment in swarmer cells is not only responsible for delayed transition from a single swirl to multi-swirls in confinement with respect to confinement size, but it also reproduces dynamic polar clusters in the dilution experiment. Future work is called upon to explore the rafting phenomenon further, including in other species of bacteria, and to investigate

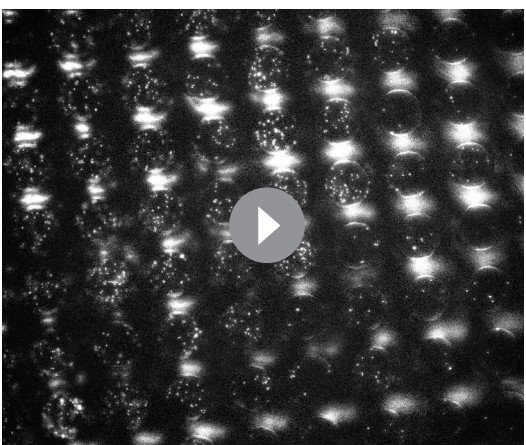

**Video 7.** Fluorescent beads' motion on DSS induced colitic mouse intestine tissue. The unidirectional rotation motion in 38 µm diameter wells indicates the presence of swarming SM3 on the tissue surface. Video was taken in 20 fps and compressed to play in 30 fps.
https://elifesciences.org/articles/64176#video7

whether or not there is a molecular basis for enhanced cell–cell alignment interaction among the swarming cells in rafts.

Our experiments on SM3 confirm the prediction made by Beppu et al. that a single vortex occurs when the confinement diameter d is smaller than a critical length *l\** (*Beppu et al., 2017*). Here, from *Figure 2C*, we found that the critical length for swarming SM3 is ~98 µm, whereas for concentrated planktonic SM3, it is ~34 µm. Interestingly, the same bacterial strain manifests different threshold sizes for the transition from single swirl to multiple swirls, corresponding to the two motility states. In particular, we were able to use this property to identify the motility types on mouse mucosal surfaces. The beads' motion is not a perfect swirl in every well on the colitic tissue because the mucosal surface is not as smooth as the agar surface. There are sags and crests on the inflamed mucosal surface due to the disrupted mucin layer (*De et al., 2021*). We conjectured that this unevenness hindered the swirl formation to a certain extent.

Indeed, intact swirl patterns were spotted only on limited locations where the mucosal surface was relatively flat. Nevertheless, capturing only a few wells where beads showed single-swirl motion was sufficient to show that swarming occurred on a mucosal surface.

Evidence of genetic and epigenetic regulation (*Daniels et al., 2004*; *Morgenstein et al., 2010*; *Tremblay and Déziel, 2010*; *Wang et al., 2004*), as well as cell morphology changes (e.g., cell elongation and hyper-flagellation), indicates that swarming is a different phenotype from swimming. Lacking comparison under the same experimental conditions, bacterial swarming might be perceived as merely a dense group of cells swimming on a surface. Here, through geometry confinement, we show that *Enterobacter* sp. SM3 swarming manifests different characteristics from swimming planktonic cells at comparably high concentration. Furthermore, we found that not only SM3, but several other species of gram-negative swarming bacteria have shown similar properties based on our limited additional experiments (see Appendix 1), which implies that strong alignment interaction among the swarmer cells might commonly exist in swarming bacteria.

The key experimental method of mounting PDMS micro-disk array on agar differentiates swarming motility from swimming motility at mesoscopic or even macroscopic scales, providing a visual assay to detect swarming behavior on agar and tissue surfaces. This study's findings provide the rationale for developing applications such as isolating bacterial swarmers from a polymicrobial environment and developing diagnostics for the presence of in vivo swarming (e.g., detecting urinary or fecal swarming bacteria in catheter infections or intestinal inflammation, respectively) (*Arikawa and Nishikawa, 2010*; *Lane et al., 2007*). Specifically, the sensitivity to confinement size indicates that a quantitative ranking system for different species of swarmers could be established based on the characteristic

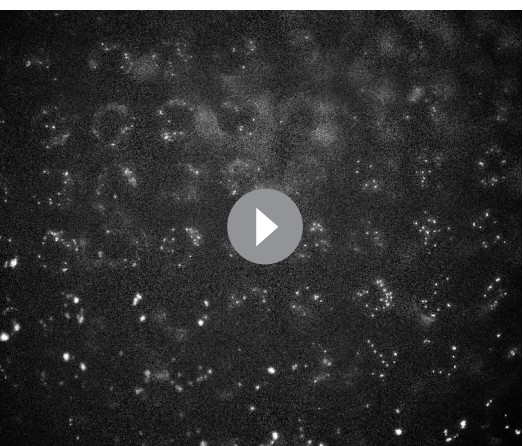

**Video 8.** Fluorescent beads' motion on normal mouse intestine tissue. The random motion in 38 µm diameter wells indicates predominantly planktonic SM3 on the normal mice tissue surface. Video was taken in 20 fps and compressed to play in 30 fps.
https://elifesciences.org/articles/64176#video8

upper bound well size that stabilizes the confined motion pattern into a single swirl. Such a ranking system may prove useful in future investigations on the implications of swarming bacteria in host physiology and pathophysiology.

# Materials and methods

**Key resources table**

| Reagent type (species) or resource | Designation | Source or reference | Identifiers | Additional information |
|---|---|---|---|---|
| Strain, strain background (*Enterobacter sp.* SM3) | Wild Type SM3 | https://doi.org/10.1101/759886 | Novel Strain | |
| Chemical compound, drug | D-mannose | Research Product International | Cas No. 3458-28-4 | |
| Chemical compound, drug | Cephalexin | Sigma-Aldrich | C4895 | |
| Chemical compound, drug | Surfactin | Sigma-Aldrich | S3523 | |
| Software, algorithm | ImageJ | NIH (https://imagej.nih.gov/ij/) | | Version: v1.59e |
| Software, algorithm | Python: DBSCAN | https://scikit-learn.org/stable/index.html | | Version: 0.24.1 |
| Software, algorithm | MATLAB | Mathworks | | Version: R2019b |
| Software, algorithm | PIVlab toolkits | Mathworks | | Version: 2.39 |
| Software, algorithm | Prism | GraphPad Software | | Version: 8.4.3 |

## PDMS confinement sheet fabrication

Polydimethylsiloxane (PDMS) microwell confinement sheets with different combinations of well sizes and depths were fabricated using a soft photolithography technique. Patterns of the confinement were first designed using the software 'L-Edit' and then uploaded into a maskless aligner (MLA 150, Heidelberg). On a 3-inch silicon wafer (University Wafer Inc), photoresist gel SQ25 (KemLab, Inc) was spin-coated at 2000 rpm (spin speed varies according to the desired coating thickness). After baking, UV exposure, and chemical development, the microwells' designed pattern was shown on the wafer (molding). Then, PDMS (Dow Corning Sylgard 184) base elastomer was mixed with the curing agent at the ratio of 10:1 in weight. The mixture was cast onto the patterned silicon wafer. Two grams of the mixture ended up with a PDMS sheet about 0.5 mm thick. The PDMS solidified at room temperature within 48 hr and the sheet was cut into pieces and peeled off from the silicon wafer before use (demolding).

## Bacterial growth and confinement (*Figure 7A*)

*Enterobacter* sp. SM3 is a novel swarming bacterial strain isolated from inflammatory mice (*De et al., 2021*). SM3 was transferred from −80°C glycerol stock to fresh LB (Lysogeny Broth: water solution with 10 g/L tryptone, 5 g/L yeast, and 5 g/L NaCl) and shaken overnight (~16 hr) in a 37°C incubator at 200 rpm. For swarming under confinement assay (*Figure 7A*, red arrows), 2 μL overnight bacterial culture was inoculated on the center of an LB agar plate (10 g/L tryptone, 5 g/L yeast, 5 g/L NaCl, and 5 g/L agar; volume = 20 mL/plate) and kept in a 37°C incubator. After the population of bacteria started swarming for 2.5 hr, a PDMS chip (~1 cm$^2$) was mounted upon the edge of the swarming colony and the Petri dish was transferred onto the microscope stage for observation. For swimming under confinement assay (*Figure 7A*, blue arrows), overnight bacterial culture was resuspended in fresh LB (1:100 in volume) and shaken in the 37°C incubator at 200 rpm for 2.5 hr. The freshly grown culture was centrifuged at 1500 g for 10 min and ~98.6% of the supernatant was removed so that the resultant cell density is about 70 times the freshly grown culture. Ten (10) μL concentrated bacteria culture was inoculated on the LB agar plate, and the PDMS chip was mounted immediately. The plate was then transferred onto the microscope stage for observation. For other bacteria strains, including *Bacillus subtilis* 3610, the procedure was the same as that of SM3. There are thousands of wells on one PDMS chip, and when mounted on a bacterial spot or colony edge, hundreds of them are infiltrated by bacteria. The PDMS chip was first brought to contact with the bacteria and then gently mounted onto the agar. By doing so, there was a cell density gradient across an array of wells, with the wells closer to the bacteria spot or colony center having higher cell density. We focused on the area where the confined bacteria showed collective motion, that is, the

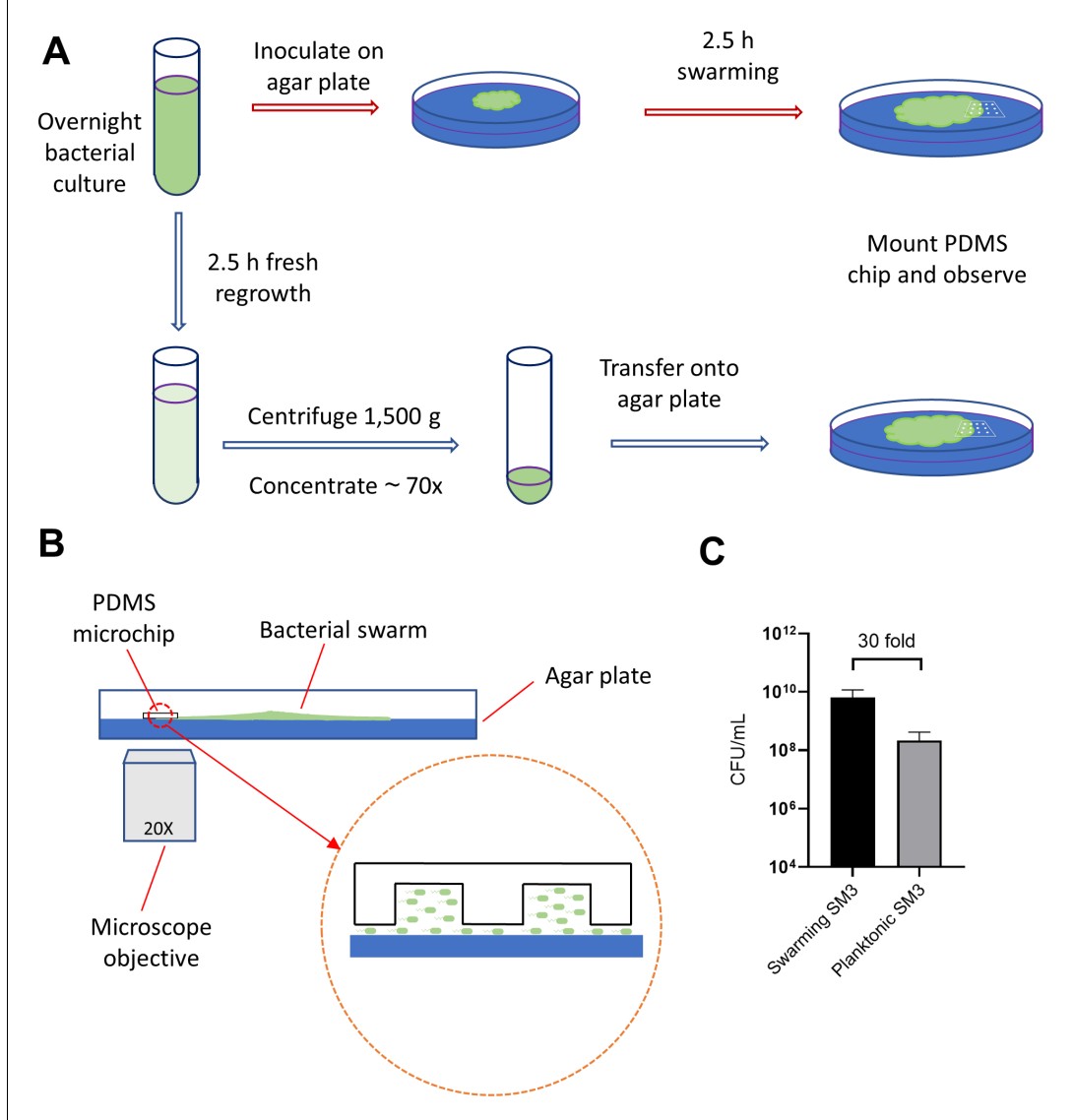

**Figure 7.** Illustration of experimental procedure. (**A**) Schematic of sample preparation procedure. Red arrows represent the assay procedure for swarming bacteria. Blue arrows represent the assay procedure for swimming planktonic bacteria. (**B**) Schematic diagram of the experimental device (side view). The gap of a few microns between the PDMS chip and the agar surface, illustrated in magnified view, allows the bacteria under the chip to spread. (**C**) Cell density measured by colony forming unit (CFU/mL) of swarming SM3 and swimming, planktonic SM3. Swarming SM3 cell density is measured after SM3 swarming on an agar surface for 2.5 hr while swimming SM3 cell density is measured for overnight SM3 culture being regrown in fresh Lysogeny Broth (LB) for 2.5 hr. Since cell density of swarming SM3 was higher than that of planktonic SM3, the latter was concentrated to acquire comparable cell density before being applied on the agar plate.

cell density was not too high to oversaturate the well, nor too low so that cells move independently from each other.

## Bacterial cell density measurement (*Figure 7B*)

Two and half hour (2.5 hr) freshly grown SM3 was subjected to different factors of dilution in LB, such as $10^2$, $10^3$, until $10^8$. Fifty (50) µL of each diluted culture was inoculated and spread on 1.5% LB agar plate (10 g/L tryptone, 5 g/L yeast, 5 g/L NaCl, and 15 g/L agar; volume = 20 mL/plate) and was incubated at 37°C for 16 hr. Bacterial colonies appeared on the agar plates and the number of colonies was counted for the dilution that resulted in the colony's number on the order of 100. The colony forming unit per microliter (CFU/mL) was calculated by dividing the colony number by the

sampled volume. For swarming SM3, the cell density was measured similarly. On the edge of the swarming colony, a chunk of swarming SM3 (~1 mm wide) was picked by an 8 mm-wide square spatulate containing a small chunk of agar at the bottom to ensure all the cells in that region were sampled. The 1 mm × 8 mm chunk of swarming SM3 was then mixed into 1 mL LB for CFU determination. The colony thickness was measured by microscopy focusing on the top of the colony and the top of the agar surface (i.e., at the bottom of the colony), keeping track of the fine adjustment knob readings. Particles of baby powder (~several micrometers in diameter) were spread on the surface of either swarm colony or agar in order to aid in the microscope focus. The thickness of the swarming colony was calculated based on the calibration of the fine adjustment knob tick readings. Then the cell density was estimated by CFU/mL. CFU was calculated for both swarming and swimming SM3 to ensure the densities of live cells in these two cases were comparable inside the wells. We consider colony-forming unit counting a better way to match the concentrations since dead cells do not contribute to the motion in the well.

## Bacterial cell length and motility

For swimming SM3, 2.5 hr freshly grown culture was diluted 100 times in LB, 50 μL of which was transferred on a glass slide and covered with a coverslip. The sample slide was placed under the microscope (Olympus CKX41, 20×), and image sequences were captured. Cell lengths were measured using ImageJ (v1.59e) freehand label tool. Cell speed was calculated as the traveling trajectory length divided by the traveling duration (~1 s). For swarming SM3, a chunk of swarming bacteria was collected from the swarming colony edge and diluted into 1 mL LB. A glass slide and a cover slip sandwiched a droplet of 50 μL mixed culture, and the rest of the procedure was the same as that for the swimming SM3.

## Swimming SM3 with different treatments
### Cephalexin treatment
Overnight SM3 culture was diluted 100 times in fresh LB and incubated in a 37°C shaker at 200 rpm for 1.5 hr. Cephalexin (CEP) (C4895; Sigma-Aldrich) was added to the culture so that the CEP's resultant concentration was 60 μg/mL. The culture was kept in the shaker for another 2 hr before use.

### Surfactin additions
After 2.5 hr regrown culture was centrifuged, excess supernatant was removed, and surfactin (S3523; Sigma-Aldrich) was added so that the resulting concentrations of surfactin were 10, 50, 100, and 500 μM in four preparations. At the same time, the cell density remained comparable to that of swarming SM3.

### Addition of swarming supernatant
Before swarming SM3 covered the plate, the colony was scratched using a PDMS spatulate (~0.5 cm²) and transferred into 1 mL deionized water. The mixture was sucked into a syringe and filtered with a 0.2 μm size filter. The solution was then lyophilized to powder and then dissolved into the concentrated planktonic SM3 of roughly the same volume as the collected swarm fluid. Thus, the concentrated planktonic SM3 was subjected to soluble compounds of the same concentration as in the supernatant of swarming SM3.

## Swarming SM3 with different treatments
### Soft scratching with PDMS
After SM3 swarmed on the agar plate for 2.5 hr, a piece of PDMS (~0.5 cm × 1 cm) was used to gently scratch the edge of the swarming colony so that the swarming cells were disturbed. A PDMS confinement chip was then mounted on the disturbed region for observation.

### Spun down in LB
After swarming for 2.5 hr, SM3 cells were collected from the colony's edge using a blotting method (*Darnton et al., 2004*). Briefly, the cells were blotted by a piece of spare PDMS (1 cm × 2.5 cm) and transferred to 1 mL LB. The swarming cells were centrifuged at 1500 g for 10 min, and LB was

removed to restore the initially high cell density. Ten (10) µL of the swarming cells thus treated were applied on a new agar surface and a PDMS confinement chip was mounted for observation.

### D-mannose
A droplet of 50 µL 0.2% (w/v) D-mannose (Cas No. 3458-28-4; RPI) was pipetted on a swarming SM3 colony edge. After 1–2 min, when the cell density became uniform again, a piece of PDMS confinement chip was applied to the D-mannose treated region for observation under a microscope.

## VOP measurement and spatial autocorrelation function
Image sequences of swarming or swimming SM3 under confinement were taken by a microscope camera (ThorLabs, Kiralux CS505MU) and then processed using a particle image velocimetry (PIV) package in MATLAB. The velocity field was marked for the confined bacteria and the VOP was calculated using the equation in *Figure 1E*. Using the velocity field information, we calculated the spatial autocorrelation function through the equation $C_r(\Delta r) = <\frac{v(r_0)\cdot v(r_0+\Delta r)}{\left|v(r_0)^2\right|}>$, where $r_0$ is the local position vector and $\Delta r$ is the displacement vector (*Patteson et al., 2018*). A Python script was written to calculate all the $C_r$ values in the region of interest (ROI) with a label of $\Delta r$ values. These $C_r$ values were then plotted as a function of $\Delta r$.

## Cluster analysis
On the swarming SM3 colony edge or concentrated swimming SM3 inoculation, a droplet of 50 µL deionized water was added via a pipette. Once the fluid flow stabilized, image sequences were captured at the diluted swarming or planktonic SM3 sample locations. In a region of 130 µm × 130 µm, the velocity field was calculated using the PIVlab toolkit, and the vectors with magnitude below 4 µm/s were removed in order to exclude non-motile bacteria. Once the moving cells were identified, a Python script was implemented to perform the clustering analysis using the function of DBSCAN (Density-Based Spatial Clustering of Applications with Noise) (*Ester et al., 1996*) where the parameter ε was set to 50, which specifies how close points should be to each other to be considered a part of a cluster, and the minimum number of points to form a cluster was set to 20.

## Numerical simulations
The numerical simulation consists of a 2D system of $N$ particles. The position $r$ of each particle is modeled via the following overdamped Langevin equation:

$$\partial_t \boldsymbol{r_i} = v_0 p_i - \sum_{j\neq i} G_\theta(d_{ex}, r_{ji}) + \sqrt{2D_T}\xi_i \tag{1}$$

It is assumed that particles are cruising at a constant speed of $v_0$ in the direction of $p_i = [\cos(\theta_i), \sin(\theta_i)]$. The second term includes the exclusion forcing term from all neighboring particles residing at a distance $r_{ji}$ closer than the exclusion range $d_{ex}$. The last term is the thermal fluctuation term with the translational diffusivity $D_T$ and a zero-mean and delta-correlated noise term $\xi$. The direction of motion $\theta_i$ of each particle is updated by the interaction terms $F_\theta$, which includes alignment, anti-alignment, and repulsion effects with all neighboring particles and the rotational diffusion term with diffusivity of $D_r$ and noise term $\zeta$:

$$\partial_t \theta_i = \sum_{j\neq i} F_\theta(\boldsymbol{r_{ji}}, p_i, p_j) + \sqrt{2D_r}\zeta_i$$

The details of the binary interaction terms $G_\theta$ and $F_\theta$ are provided in Appendix 2. The simulation starts with random initial position and orientations, followed by numerical integration of equations (1) and (2) using a first-order Euler method. The integration time step $\Delta t$ is chosen small enough to ensure numerical stability and independence of long-term dynamics from the time step increment. The interaction of particles with a circular boundary is modeled through a reflective boundary condition. The particles are reflected off the boundary with an angle equal to their incident angle. In the diluted cases, the reflecting solid boundary is replaced with a periodic boundary condition to avoid the effect of boundary scattering on the dynamics in a bulk sample, which was observed under a microscope.

# Detecting bacterial motility on mouse intestinal mucosal tissue using PDMS chips

Six-week-old female C57BL/6 mice (Jackson Laboratories, Bar Harbor, ME; #000664) were administered 3%(w/v) DSS (MPBio; # 160110) in animal facility drinking water daily to induce acute colitis (*De et al., 2021*). After 9–12 days, when the average weight loss reached 20%, mice were euthanized using isoflurane anesthesia and large intestines were harvested. For controls, conventional 6-week-old female C57BL/6 mice exposed to drinking water with DSS-vehicle added were also sacrificed and the intestines were collected. This study was approved by the Institute of Animal Studies at the Albert Einstein College of Medicine, Inc (IACUC # 20160706 and 00001172). Intestinal tissue was surgically exposed, cleansed with 35% (v/v) ethanol, and rinsed with PBS twice. The mucosal surface of the tissue was cultured (on agar streaks) for any residual bacteria and only used when there were no bacterial colonies on aerobic or anaerobic cultures. Prior to experiments, a portion of the mucosal tissue was also harvested after ethanol cleansing for histology and to validate its histologic integrity with respect to non-cleansed DSS-exposed tissue. Tissues were spread on a 1% agar plate with the inner side facing up, and overnight SM3 bacterial culture was inoculated on one end of the tissue. The agar plate was incubated under 37°C for 4.5 hr to allow SM3 bacteria to duplicate and move on the tissue surface. PDMS chips (d = 38 µm) were coated with 0.5 µm fluorescent beads (Dragon green; Bangs Laboratory, IN) and cut into strips to fit the tissue's size. The PDMS strip was mounted on and covered the tissue surface. Bead motion was observed under the fluorescence microscope (Olympus CKX41) with a 20× objective.

## Acknowledgements

We thank Daniel B Kearns from Indiana University at Bloomington for providing the *Bacillus subtilis* 3610 bacteria strain, and Cori Bargmann at Rockefeller University for gifting us the bacteria strain *Serratia marcescens* Db10. We thank Hui Ma for his assistance in the cleanroom and discussion of the work. NIH Grants R01CA222469 and ES030197 supported this work. HK acknowledges support in the form of a Leon N Cooper Postdoctoral Fellowship.

## Additional information

### Competing interests

Weijie Chen: Weijie Chen, Neha Mani, Jay X. Tang, and Sridhar Mani filed a U.S. patent application (Application No. 63033369). Neha Mani, Sridhar Mani, Jay X Tang: Weijie Chen, Neha Mani, Jay X Tang, and Sridhar Mani filed a U.S. patent application (Application No. 63033369). The other authors declare that no competing interests exist.

### Funding

| Funder | Grant reference number | Author |
|---|---|---|
| National Institutes of Health | R01CA222469 | Sridhar Mani |
| National Institutes of Health | ES030197 | Sridhar Mani |

The funders had no role in study design, data collection and interpretation, or the decision to submit the work for publication.

### Author contributions

Weijie Chen, Conceptualization, Data curation, Software, Formal analysis, Validation, Investigation, Visualization, Methodology, Writing - original draft; Neha Mani, Conceptualization, Software, Formal analysis, Validation, Investigation, Visualization, Methodology; Hamid Karani, Data curation, Software, Formal analysis, Validation, Methodology, Writing - original draft; Hao Li, Resources, Methodology; Sridhar Mani, Conceptualization, Supervision, Funding acquisition, Validation, Investigation, Methodology, Project administration, Writing - review and editing; Jay X Tang, Conceptualization,

Resources, Supervision, Validation, Investigation, Methodology, Project administration, Writing - review and editing

## Author ORCIDs
Weijie Chen (iD) https://orcid.org/0000-0002-7105-5645
Jay X Tang (iD) https://orcid.org/0000-0002-1022-4337

## Ethics

Animal experimentation: The animal tissue samples were acquired from Albert Einstein College of Medicine in strict accordance with the recommendations in the Guide for the Care and Use of Laboratory Animals of the National Institutes of Health. This study was approved by the Institute of Animal Studies at the Albert Einstein College of Medicine, Inc (IACUC # 20160706 & 00001172).

## Decision letter and Author response
Decision letter https://doi.org/10.7554/eLife.64176.sa1
Author response https://doi.org/10.7554/eLife.64176.sa2

## Additional files
### Supplementary files
- Source code 1. Main program of the simulation with initial inputs of the parameters.
- Source code 2. Functions called in the main program that describe particle interactions.
- Transparent reporting form

### Data availability
All data generated or analyzed during this study are included in the manuscript and supporting files. Source code files have been provided for Figure 5.

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

## Appendix 1

### Other swarming bacteria show similar behavior to SM3

We also tested *Enterobacter* sp. SM1, a slower swarming strain nearly identical to SM3 at the genetic level (*De et al., 2021*), as well as other species of bacteria such as *Serratia marcescens* (including one lab strain Db10 and another strain, H3, isolated from a human patient), *Citrobacter koseri* H6, and *Bacillus subtilis* 3610. All the tested strains, with the exception of *B. subtilis*, showed similar motion pattern divergence between swimming planktonic cells and swarming cells when compared in confinement as shown for SM3 (*Appendix 1—figure 1A*). SM1, H6, and H3 all behave like SM3, that is, they all showed clusters of cells moving in packs when the swarming colony was diluted and uniformly dispersed when the concentrated planktonic cells were diluted (data not shown).

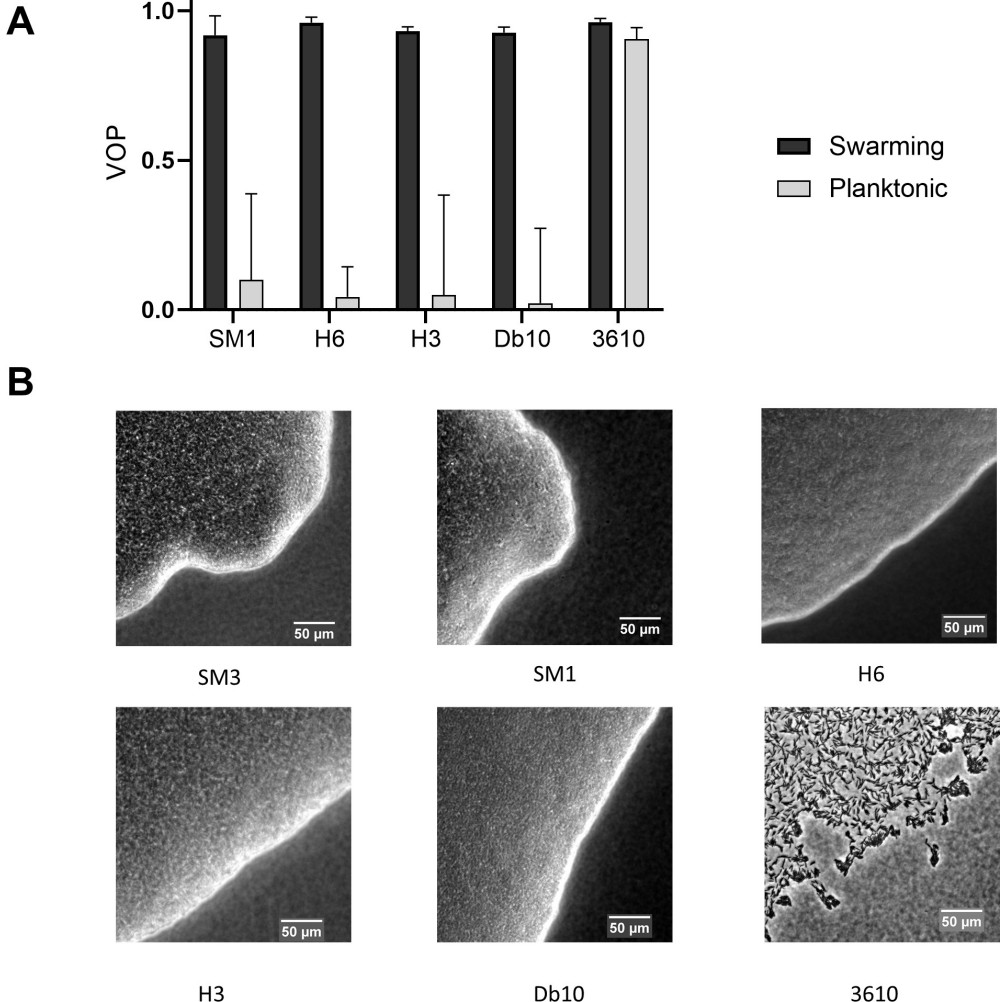

**Appendix 1—figure 1.** Comparisons of vortex order parameter (VOP) under confinement and swarm front among several species of bacteria. (**A**) VOP of concentrated planktonic and swarming *Enterobacter* sp. SM1, *Citrobacter koseri* (H6), *Serratia marcescens* (H3), *Serratia marcescens* (Db10), and *Bacillus subtilis 3610* confined in the PDMS microwells of 58 µm in diameter and 22 µm in depth. The bars indicate averages with standard deviation (+ SD) over five microwells. (**B**) Swarm front of the tested bacteria. *B. subtilis* 3610 forms a monolayer, loose swarming colony whereas all the other bacterial strains form multilayer, compact swarming colonies.

One notable exception is *Bacillus subtilis*. Swarming and concentrated planktonic *Bacillus subtilis* 3610 show the same motion pattern across different confinement sizes. For well diameter d ≤ 90 µm, both swarming and swimming *B. subtilis* form single swirls while for well diameter d ≥ 112 µm, they both break into mesoscale vortices. *B. subtilis* is a gram-positive bacterium, whereas SM3, SM1,

H6, H3, and Db10 are all gram-negative. We speculate that swarming *B. subtilis* does not have either as strong cell–cell interaction as SM3 and its gram-negative cohort we tested, or as significantly enriched extracellular matrix materials to sustain stronger cell–cell alignment in its swarm state as among its planktonic swimmers when concentrated to comparable concentration. Indeed, the interaction is not so different between the swarming and planktonic *B. subtilis* 3610 cells since we found the diluted swarming cells to disperse uniformly, and with no clustering behavior, much like diluted planktonic cells. The swarming colony thickness for *B. subtilis* also differs from the other strains. It is known that swarming *B. subtilis* produces abundant surfactant, resulting in a widespread, monolayer, non-compact colony (*Be'er and Ariel, 2019*; *Jeckel et al., 2019*). In contrast, swarming SM3 and the other tested bacteria all form multilayer colonies that are as thick as 20–40 μm at the edge (*Appendix 1—figure 1B*). The thickness of SM3 swarm and that of its gram-negative cohort on agar enabled them to extend strong cell–cell alignment through the entire depth of PDMS wells of 22 μm depth and of disk diameters between 40 and 90 μm. In contrast, the cell–cell alignment is notably weaker among planktonic cells of comparable concentration.

## Appendix 2

### Mathematical modeling and computer simulation
#### A simplified treatment of swarming bacteria

Most particle-based models for self-propelled microswimmers incorporate detailed hydrodynamics of elongated rods in a low Reynolds number environment (*Costanzo et al., 2012*; *Lushi and Peskin, 2013*; *Lushi et al., 2014*; *Saintillan and Shelley, 2007*). However, the dynamics of bacterial swarming comprise a complex interplay between several physical and chemical interactions that go beyond hydrodynamic and steric effects. Cell interactions with the extracellular polymeric network, mechanical locking and intertwining of flagella including formation of intercellular flagellar bundles between adjacent swimming cells (*Copeland and Weibel, 2009*; *Kearns, 2010*) are a few examples whose underlying mechanisms are not fully understood. In the absence of a comprehensive model that captures many interactions among swarming bacteria, we seek a simplified description of active particles interacting via competing interactions that capture the essential dynamics of both swarming and planktonic bacteria. Our focused aim in connection with the experimental study in this report is to discern the distinct, collective behaviors of swarming bacteria from their planktonic, swimming counterpart, in comparable concentration, and under the same extent of spatial confinement.

There are numerous approaches for incorporating the relevant physical interactions between active particles (*Grossmann et al., 2014*; *Grossmann et al., 2015*; *Wensink et al., 2012*; *Wensink and Löwen, 2012*) (readers are referred to Bär et al. for a recent review (*Bar et al., 2020*), for example, on models for dry and wet interacting self-propelled rods). Here, we choose the binary interaction model introduced by Großmann et al. (*Grossmann et al., 2014*; *Grossmann et al., 2015*) based on the fact that hydrodynamic couplings among the swimmers can induce both alignment and anti-alignment effects (*Baskaran and Marchetti, 2009*). The simplified model we employ also allows us to implicitly embed unknown interactions of cells with extracellular polymeric network and possibly, mechanical locking of flagella between adjacent cells in alignment, anti-alignment, and repulsion torque terms.

### Numerical model and simulation

The dynamics of $N$ interacting active particles have been modeled in a two-dimensional space using the overdamped Langevin-based equations, assuming that inertia is negligible in a low Reynolds number environment. The position $r$ and orientation $\theta$ of particle $i$ are calculated using the following stochastic differential equations:

$$\partial_t \mathbf{r_i} = v_0 \widehat{p}_i - \sum_{j \neq i} k_{ex} r_{ji} H(d_{ex} - r_{ji}) + \sqrt{2D_T} \xi_i \tag{1}$$

$$\partial_t \theta_i = \sum_{j \neq i} F_\theta \left( \mathbf{r_{ji}}, \widehat{p}_i, \widehat{p}_j + \sqrt{2D_r} \varsigma_i \right) \tag{2}$$

In *Equation (1)*, the particles' self-propulsion speed is set to be a constant $v_0$. The swimming direction of particle $i$ points along $p_i = [\cos(\theta_i), \sin(\theta_i)]$. This simple assumption is based on our experimental observations, suggesting that the bacterial velocity in the suspension is largely independent of the local cell density. The second term incorporates the central exclusion force term with a spring constant $k_{ex}$, which acts over the relative distance $r_{ji}$ with all the neighboring particles $j$. This exclusion force term applies only when $r_{ji}$ gets smaller than the exclusion range $d_{ex}$ (represented as a Heaviside function H). The last term in *Equation (1)* is the Brownian fluctuation term with the corresponding translational diffusivity $D_T$, where $\xi_i$ is the white noise with zero mean and correlation $\delta(t)$ as a delta function.

Two terms influence the temporal change in the orientation of each particle. The first term on the right-hand side of *Equation (2)* includes all the binary interaction terms. The last term on the right-hand side of *Equation (2)* is the contribution from the angular Brownian fluctuation with the rotational diffusion $D_r$ and a zero mean and delta-correlated stochastic white noise $\varsigma$. In the present study, we employ the pair-wise interaction model introduced by Grossman and co-workers (*Grossmann et al., 2014*; *Grossmann et al., 2015*), which successfully reproduces various

macroscopic patterns that occur in dense bacterial suspensions. The pair-wise interaction term is based on a zonal model (illustrated in *Appendix 2—figure 1* below), capturing the alignment, anti-alignment, and repulsion effects. It is formulated in the following form (*Grossmann et al., 2014*; *Grossmann et al., 2015*):

$$F_\theta\left(\boldsymbol{r_{ji}}, \hat{p}_i, \hat{p}_j\right) = k_r \mathfrak{H}\left(r_r - r_{ji}\right)\sin\left(\theta_i - \theta_{ji}\right) + \mu\sin\left(\theta_j - \theta_i\right) \tag{3}$$

$k_r$ is the magnitude of the constant repulsion interaction that applies over a distance of $r_r$ around the particle (*Appendix 2—figure 1*). The second term in *Equation (3)* represents the alignment and anti-alignment effects. The magnitude of the aligning interaction $\mu$ is distance-dependent and is defined as (*Grossmann et al., 2014*; *Grossmann et al., 2015*):

$$\mu = \begin{cases} \mu^+\left(1 - \left(r_{ji}/r_a\right)^2\right) & 0 \le r_{ji} \le r_a \\ -\mu^- \dfrac{4\left(r_{ji-r_a}\left(r_{aa} - r_{ji}\right)\right)}{\left(r_{aa} - r_a\right)^2} & r_a \le r_{ji} \le r_{aa} \end{cases} \tag{4}$$

where $\mu^+$ and $\mu^-$ are the strengths, and $r_a$ and $r_{aa}$ are the ranges of alignment and anti-alignment interactions, respectively.

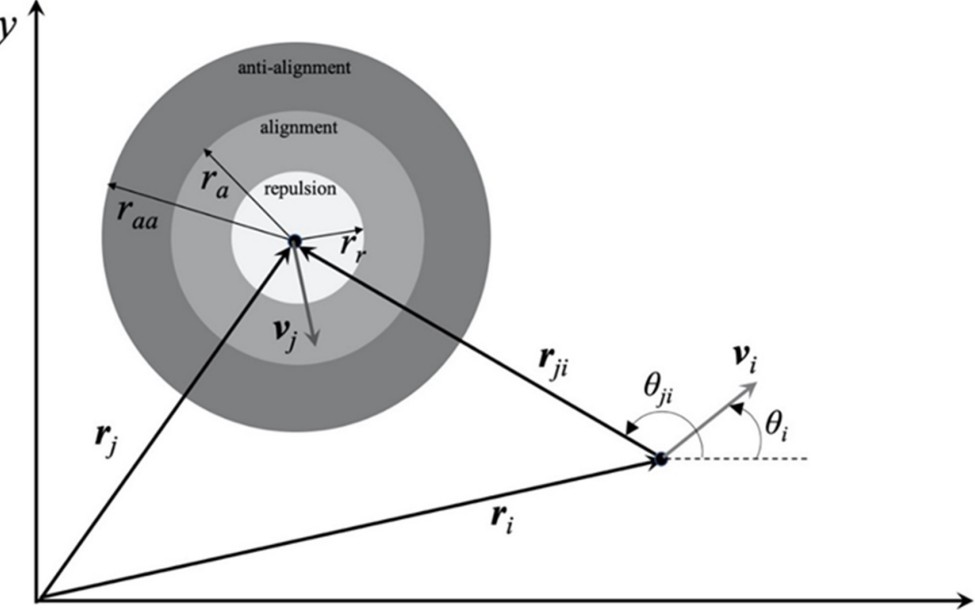

**Appendix 2—figure 1.** Schematic of the zonal pair-wise interaction model showing anti-alignment, alignment, and repulsion zones with the corresponding interaction radii $r_{aa}$, $r_a$, and $r_r$.

We numerically integrate *Equations (1) and (2)* using the first-order Euler scheme. Initially, the particles are randomly distributed with random orientations. The integration time step $\Delta t$ is selected sufficiently small to ensure both numerical stability and also independence of long-term statistics from $\Delta t$. The simulation time is set long enough to let the system reach a dynamic steady state. The interaction of particles with the bounded circular domain is modeled via a reflective boundary condition.

## Assessment of simulation parameters

Swarming cells secrete large amounts of surface-active compounds that modify the surface tension locally (*Fauvart et al., 2012*; *Ke et al., 2015*), as well as micro-viscosity of the fluid (*Copeland and Weibel, 2009*), which along with the formation of intercellular flagellar bundles between neighboring cells can enhance the cohesive interaction and alignment among swarmer cells. Thus, simulation parameters must be chosen to capture different behaviors between the planktonic and swarmer cells.

Two different sets of interaction parameters have been used to differentiate the swarming and planktonic cases, and these parameters are summarized in *Appendix 2—table 1*. The correlation and anti-correlation lengths in the experimental velocity auto-correlation plots shown in *Figure 2D and E* serve as a guide for the corresponding alignment and anti-alignment ranges in simulations. We set the exclusion parameters $k_{ex}$ and $d_{ex}$ to fixed values of 0.02 and 3.5 µm, respectively. The exclusion range $d_{ex}$ of 3.5 µm approximately accounts for the finite size effect, that is, effective distance around a point-like particle that, along with repulsion force $k_{ex}$, guarantees in our simulations volume exclusion in a region approximately defined by the actual size of the flagellated bacteria.

**Appendix 2—table 1.** Simulation parameters used for swarming and planktonic cases.

|  |  | Swarming | Planktonic |
| --- | --- | --- | --- |
| Repulsion | $k_r$ (rad/s) | 5 | 5 |
|  | $r_r$ (µm) | 6 | 6 |
| Alignment | $\mu^+$ (rad/s) | 0.3 | 0.3 |
|  | $r_a$ (µm) | 20 | 15 |
| Anti-alignment | $\mu^-$ (rad/s) | 0.1 | 0.1 |
|  | $r_{aa}$ (µm) | 30 | 30 |

In simulations, we define a 2D particle density as $\rho = N/A_{dom}$, where $N$ is the number of particles, and $A_{dom}$ is the simulation domain area in the units of µm$^2$. The confinement sizes are similar to microwell diameters used in the experiments. The number of particles $N$ is estimated based on the experimental knowledge that there were roughly 1000 cells in a cylindrical microwell of 74 µm diameter and 22 µm well depth. This first leads to an approximate cell–cell spacing of 4.5 µm in 3D, so that the well can be approximated as consisting of five layers of cells over the disk depth. The projected 2D density is then estimated to be one cell per 4.5 µm × 4.5 µm area, so that $\rho = 0.05$ µm$^{-2}$. In the unconfined dilute case, we set $\rho = 0.025$ µm$^{-2}$ and we replace the bounded domain with a periodic boundary in order to further simulate unconfined condition. In all simulations, we ignore translational diffusion as the Péclet number Pé = $v_0 l/D_T \gg 1$, where $v_0$ is the average swimming speed and $l$ is the effective length of bacteria (several µm, including flagella). It is further assumed that particles experience a rotational diffusion $D_r$ of $\approx$ 0.3 (rad$^2$/s), which corresponds to that of a spherical particle with an effective radius $r \approx 0.85$ µm in water and at room temperature. The results are independent from noise level up to $D_r \approx 0.9$ (rad$^2$/s), beyond which rotational diffusion would suppress alignment of neighboring particles and would break single- or multi-swirl patterns into gas-like state.

The simulation results at high particle density $\rho = 0.05$ µm$^{-2}$ for a few representative confinement sizes are shown in *Appendix 2—figure 2*. As *Appendix 2—figure 2* illustrates, the macroscopic behavior of both swarming and planktonic cells is affected by the confinement size. The corresponding change in vortex order parameter (VOP) marks the transition from a single vortex to multiple swirls. Compared to the swarming case, the lower value of alignment range in the planktonic case triggers an onset of the transition at smaller disk diameter by a factor of 2–3 (*Appendix 2—figure 2*).

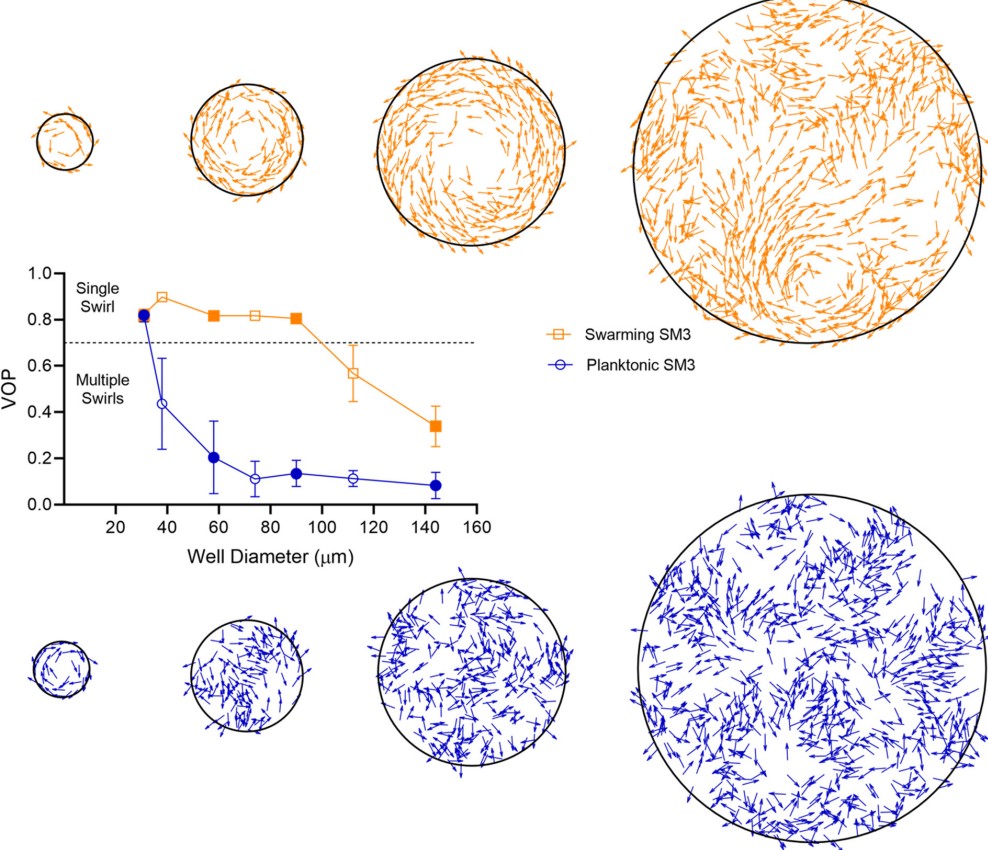

**Appendix 2—figure 2.** Representative patterns at different sizes of the bounded domain. Top row: Swarming; Bottom row: Planktonic. The corresponding domain sizes and VOP values are marked as filled symbols. The particle density is kept constant as the area of the simulated region increases. Simulation parameters are based on the values summarized in *Appendix 2—table 1*.

The set of simulation parameters in *Appendix 2—table 1* predicates that alignment interactions in planktonic cells are suppressed via a lower alignment range of 15 μm compared to 20 μm in swarmer cells. This implies that swarmer cells tend to align with their neighboring swimming cells up to a larger distance than planktonic cells. The simulation results provide valuable physical insights as the patterns predicted closely resemble the experimental observation. More advanced real-time visualization of bundling dynamics in swarmer cells (*Copeland and Weibel, 2009*), along with biochemical characterization of the bacterial fluids, and the micro-rheology measurements within local, extracellular polymeric network (*Guadayol et al., 2020*) will be required in order to shed light on the underlying nature of enhanced alignment interaction in swarmer cells. These properties rely on experimental efforts beyond the scope of this report. If determined, they will facilitate the development of more comprehensive particle-based models.

