## [Decision Letter]

**Acceptance summary:**

In this paper, the authors report new results on the collective behavior exhibited by bacteria under confinement. Using microwells of specific sizes on agar surfaces, they found swarming bacteria exhibit a "single-swirl" motion pattern and concentrated planktonic bacteria exhibit "multi-swirls" motion pattern in the diameter range of 31-90 μm. Systematic experiments explore parameters defining the divergence of motion patterns in confinement including cell density, cell length, cell speed and surfactant. They conclude that the single-swirl pattern depends on cohesive cell-cell interaction mediated by biochemical factors removable through matrix dilution.

**Decision letter after peer review:**

Thank you for submitting your article "Confinement Discerns Swarmers from Planktonic Bacteria" for consideration by *eLife*. Your article has been reviewed by 2 peer reviewers, and the evaluation has been overseen by a Reviewing Editor and Aleksandra Walczak as the Senior Editor. The following individual involved in review of your submission has agreed to reveal their identity: Yan He (Reviewer #1).

The reviewers have discussed the reviews with one another and the Reviewing Editor has drafted this decision to help you prepare a revised submission.

As the editors have judged that your manuscript is of interest, but as described below that major revisions are needed before it can be accepted for publication, we would like to draw your attention to changes in our revision policy that we have made in response to COVID-19 (https://elifesciences.org/articles/57162). First, because many researchers have temporarily lost access to the labs, we will give authors as much time as they need to submit revised manuscripts. We are also offering, if you choose, to post the manuscript to bioRxiv (if it is not already there) along with this decision letter and a formal designation that the manuscript is "in revision at *eLife*". Please let us know if you would like to pursue this option. (If your work is more suitable for medRxiv, you will need to post the preprint yourself, as the mechanisms for us to do so are still in development.)

Summary:

In this paper, the authors proposed a new approach by mounting a PDMS microwells of specific sizes on agar surface to confine swarming and planktonic SM3 cells. They found swarming bacteria exhibit a "single-swirl" motion pattern and concentrated planktonic bacteria exhibit "multi-swirls" motion pattern in the diameter range of 31-90 μm. The phase diagram shows that in smaller wells concentrated planktonic SM3 forms a single vortex and in larger wells swarming SM3 also breaks into mesoscale vortices.

In addition, they conducted systematic experiments to explore parameters defining the divergence of motion patterns in confinement including cell density, cell length, cell speed and surfactant. They concluded that the single-swirl pattern depends on cohesive cell-cell interaction mediated by biochemical factors removable through matrix dilution.

This paper gives a new method to discern swarmers from planktonic bacteria and carefully studies the factors that influence the formation of bacterial vortices under restriction.

Essential revisions:

While generally supportive of the paper, the reviewers found that major revisions are necessary to clarify aspects of the experiments and the connection between experimental findings and numerical simulations, the modeling approach and the interpretation of numerical results.

1. When the authors put the PDMS chip mounting on the edge of the swarming colony, the PDMS chip is completely attached to agar or suspended in a bacterial solution. The distance between PDMS chip and agar surface should be quantified. It is better to have a schematic diagram of the experimental device.

2. Are the bacteria still expanding outward after a PDMS chip was mounted on agar surface? The effect of PDMS chips on the expansion of bacteria on the agar surface needs to be discussed.

3. "Diluted swarming SM3 show unique dynamic clustering patterns". In the diluted bacteria experiment, the authors found that the diluted swarming bacteria can form bacterial rafts and the concentrated planktonic SM3 disperse uniformly and move randomly. Hence, when bacteria expand and gradually fill up new empty microwells, is there a process of transition from raft to single vortex state?

4. In the experiment of altering the conditions of swarming SM3, the authors diluted the swarming cells in Lysogenic Broth (LB) by 20-fold, re-concentrated the cells by centrifugation and removed extra LB to recover the initial cell density. After these operations, they found the previous single swirl turned to multiple swirls and concluded that matrix dilution can affect single-swirl pattern. The authors conjecture that centrifugation may wash away some surrounding matrix or polymers on the surface of bacteria. Therefore, the steps of centrifugation need to be presented and the effect of centrifugation on the physiological behavior of bacteria should be discussed.

5. This work covers the PDMS chip directly on the agar surface and finds that swarm and planktonic bacteria have different spatial correlation scales in the restricted microwells. The authors have done many experiments to prove the difference between clusters and planktonic bacteria and explain the reason for the single vortex. However, the conclusion is not clear. Therefore, the authors should focus more on the analysis of this new experimental phenomenon, such as critical length and vortex phase diagram, rather than just describing the experiments they did.

6. The authors mentioned the critical length for swarming SM3 is ~49 μm, whereas, for concentrated planktonic SM3, it is ~ 17 μm. Does this quoted data match that which they obtain from their experimental method? I do not see any follow-up discussion and evidence.

7. As shown in Figure 1 and Video S1 mp4, the direction of the single vortex motion of bacteria is clockwise. However, the article simply ignores that the single vortexes of bacteria all present the same direction, and there is no analysis and reasonable explanation on the vortex direction. As shown in Video S5 mp4 on the numerical simulations of circularly confined SM3, simulated bacteria vortex counterclockwise in completely opposite directions. The influence of the microwell boundary on the direction of the vortex should be clearly explained at the level of bacterial movement and preferentially with a numerical simulation.

8. Swarming and concentrated planktonic *Bacillus subtilis* 3610 show the same motion pattern across different confinement sizes. However, the authors did not give definitive conclusions and evidence. As shown in Figure S1, *Bacillus subtilis* 3610 show completely different cluster behavior. Therefore, the discussion of 3601WT may cause readers' confusion on the article. It may be better to put it in the supporting material.

9. A central finding of the present study is that the number of vortices/swirls as a function of the well diameter differs for swarming vs. swimming bacteria. The authors argue and show experimentally (Figure 2) that the behavior is identical for small and large diameters. For intermediate values, however, they report that a single swirl is observed for swarming bacteria whereas swimming bacteria show multiple swirls.

The fact that the behavior is identical for large wells suggests that the bulk behavior is identical. This is also confirmed by Figure 2E which shows that the spatial correlation function of the velocity is identical in large wells. That suggests that the boundary conditions play a central role in understanding how the observed phenomenology emerges. [Indeed, it was shown in the past that the interaction of bacteria with boundaries crucially determines the formation of swirls in confinement (Lushi, Wieland and Goldstein PNAS 111 9733 2014). The authors of this work assume reflecting boundary conditions, which – to my knowledge – contradicts the finding of Lushi et al.].

The authors, however, explain the difference of the observed patterns within their modeling study in a different way, namely by a different strength of the (anti-)alignment interactions. Changing the interaction at the level of individual cells will, however, change the bulk behavior too. Accordingly, the numerically observed bulk behavior (Figure 5B ) is very different in both cases (at a qualitative level). It is difficult to judge the difference in detail because the correlation function was not calculated for the simulations.

In short:

– The model (Figure 5A) reproduces the experimental results partially (Figure 2C), but the modeling analogue to Figure 2E is missing;

– The line of arguments seems not to be entirely consistent.

10. Inferring the interactions of active particles from observations of the emergent patterns is a highly non-trivial task. In view of this the reviewers are not entirely convinced by the arguments put forward by the authors that "more substantial cell-cell cohesive interaction(s)" are the reason why the swirling patterns formed by swarming/swimming bacteria differ.

In this context, we call attention to Ref. [Peruani, Deutsch and Bär: Phys. Rev. E 74 030904(R) 2006]. In this work, a clustering transition of self-propelled rods was described. "Rafts", referred to as clusters by Peruani et al., are observed as the aspect ratio of rods is increased. Notably, a kinetic transition towards clustering can emerge even in the absence of any attractive interactions.

In short, the observation that cells move in parallel (polar clusters) next to each other does not allow to conclude that cohesive interactions are present.

The Videos S3 and S4 provided by the authors show that the particle shape of swarming and swimming particles is clearly different. In particular, the elongated swarming bacteria show pronounced clusters (Video S3) whereas the shorter planktonic cells (Video S4) do not. The difference in aspect ratio does indeed suggest that swarming and swimming bacteria differ in their alignment interaction. However, this contradicts the observation that spatial correlations in large wells are indistinguishable (see comment 1 and Figure 2E).

Side remark: in the main text, the authors argue that changes of the aspect ratio are not the reason for an increased alignment interaction, however, in the Discussion section cell morphology changes (e.g. cell elongation and hyper-flagellation) are mentioned as an indicator that swarming is a different phenotype from swimming.

---

## [Author Response]

Essential revisions:While generally supportive of the paper, the reviewers found that major revisions are necessary to clarify aspects of the experiments and the connection between experimental findings and numerical simulations, the modeling approach and the interpretation of numerical results.1. When the authors put the PDMS chip mounting on the edge of the swarming colony, the PDMS chip is completely attached to agar or suspended in a bacterial solution. The distance between PDMS chip and agar surface should be quantified. It is better to have a schematic diagram of the experimental device.

The PDMS chip is suspended in solution. There is a thin film of solution between the chip and the agar surface. The distance between the PDMS chip is typically a few microns and that varies depending on the swarm colony thickness. A schematic diagram of the experimental device has been added to the revised Figure 7B. We thank the reviewer for this valuable suggestion.

2. Are the bacteria still expanding outward after a PDMS chip was mounted on agar surface? The effect of PDMS chips on the expansion of bacteria on the agar surface needs to be discussed.

Yes, they are. We have added the clarification in Figure 7B legend. The PDMS chip is compatible with bacterial swarming and we find that bacteria beneath the chip keep expanding and filling up new empty microwells. Indeed, bacterial swarming under a PDMS sheet has been observed previously by the Berg Lab (Turner, Zhang, Darnton, and Berg, 2010). We have included reference to that prior work.

3. "Diluted swarming SM3 show unique dynamic clustering patterns". In the diluted bacteria experiment, the authors found that the diluted swarming bacteria can form bacterial rafts and the concentrated planktonic SM3 disperse uniformly and move randomly. Hence, when bacteria expand and gradually fill up new empty microwells, is there a process of transition from raft to single vortex state?

We did not observe rafts in the microwells. When the PDMS chip was mounted on the swarm edge, an area of the microwells initially did not have bacteria in contact. It was above the agar surface directly. These empty microwells were up with water due to capillary effect within minutes. The swarm front continually expanded and entered new microwells. We repeated the experiments and observed the filling process in search of rafts the reviewer was wondering about. Initially, only a few cells entered the empty microwell and they dispersed rapidly. As more swarmers came in, some of them tended to swim along the circular boundary of the microwell and others swam freely in the bulk. When the cell density kept increasing, collective motion showed up (initially as 2-3 unstable vortices). Finally, the dense swarmers stabilized into a single-swirl motion pattern. The whole process took place in a few minutes. We conjecture that rafts do not form in the microwell but only in open space maybe because the confinement diameter is comparable to the size of the raft and the circular boundary forces the bacteria to form transient vortices instead of rafts.

4. In the experiment of altering the conditions of swarming SM3, the authors diluted the swarming cells in Lysogenic Broth (LB) by 20-fold, re-concentrated the cells by centrifugation and removed extra LB to recover the initial cell density. After these operations, they found the previous single swirl turned to multiple swirls and concluded that matrix dilution can affect single swirl pattern. The authors conjecture that centrifugation may wash away some surrounding matrix or polymers on the surface of bacteria. Therefore, the steps of centrifugation need to be presented and the effect of centrifugation on the physiological behavior of bacteria should be discussed.

The steps of centrifugation are described in the “Methods” part (Line 404-407). A discussion of the effect of centrifugation on the physiological behavior of bacteria has now been added to the manuscript as “This process is expected to wash away some extracellular matrix polymers, including perhaps those weakly adhered on the bacterial surface but would unbind upon dilution.” (Line 147-148)

5. This work covers the PDMS chip directly on the agar surface and finds that swarm and planktonic bacteria have different spatial correlation scales in the restricted microwells. The authors have done many experiments to prove the difference between clusters and planktonic bacteria and explain the reason for the single vortex. However, the conclusion is not clear. Therefore, the authors should focus more on the analysis of this new experimental phenomenon, such as critical length and vortex phase diagram, rather than just describing the experiments they did.

Taking other valuable comments by both reviewers into consideration, We have made substantial changes in the results and discussion part to quantify the experimental phenomenon. Specifically, we discussed that swarming SM3 have a larger characteristic length than the planktonic cells which is an important distinction between patterns formed by these two cell types. This new insight helped us revise the numerical simulation and confirm that the alignment range holds the key for the larger critical length of swarming SM3 and a delayed transition to a multi-swirl motion pattern in the vortex phase diagram. In addition, we also provided a mechanistic explanation on the emergence of the biased chirality of the single swirl motion pattern in confinement.

6. The authors mentioned the critical length for swarming SM3 is ~ 49 μm, whereas, for concentrated planktonic SM3, it is ~ 17 μm. Does this quoted data match that which they obtain from their experimental method? I do not see any follow-up discussion and evidence.

Sorry for the confusion. The critical size of single swirl to multi-swirl transition for SM3 were obtained from the experiments. According to Figure 2C, the critical diameters were determined as the VOP data plot in the experimental phase diagram crosses the dotted line “VOP = 0.7” (Figure 2C). For swarming SM3, it is ~ 98 μm, whereas for planktonic SM3, it is ~ 34 μm. We initially quoted radius rather than diameter in the manuscript. Now, it has been corrected to diameter so that readers are clearer when we refer those obtained from experiments (see Line 283 – 285).

7. As shown in Figure 1 and Video S1 mp4, the direction of the single vortex motion of bacteria is clockwise. However, the article simply ignores that the single vortexes of bacteria all present the same direction, and there is no analysis and reasonable explanation on the vortex direction. As shown in Video S5 mp4 on the numerical simulations of circularly confined SM3, simulated bacteria vortex counterclockwise in completely opposite directions. The influence of the microwell boundary on the direction of the vortex should be clearly explained at the level of bacterial movement and preferentially with a numerical simulation.

Both clockwise and counterclockwise rotation directions were seen for the single vortexes of SM3. However, it is in favor of clockwise direction. In one experiment, we counted the number of single swirls with either direction. In a total of 99 single swirls, 85 of them were clockwise and 14 of them were counterclockwise. We interpret this bias of swirl direction as a consequence of flagellar handedness based on our experience from previous studies, including a recent work on the same bacterium (Araujo, Chen, Mani, and Tang, 2019). When confined between the agar and PDMS surfaces, bacteria tend to swim closer to the porous agar surface (DiLuzio et al., 2005). For bacteria swimming near the agar surface, the rotating cell body experiences a drag force opposite to the rotating direction, which results in torque on the bacteria to turn right (Araujo et al., 2019; DiLuzio et al., 2005). In our case, the agar surface is the bottom surface, and the bacteria tend to have a clockwise curved trajectory which breaks the symmetry of the global motion pattern to in favor of clockwise swirl (look from the top). This symmetry breaking may not occur if we replace the agar surface with a glass surface or confine bacteria within two glass slides because in that case the effects of the upper and lower surface will cancel out. A brief discussion has been added to the manuscript as a separate paragraph (Line 236 – 249). In simulations, however, clockwise and counter-clockwise vortexed occur with equal probability. Our simulation effort mainly focuses on investigating the key factors leading to a VOP difference and the corresponding velocity auto-correlations, not the swirl direction, thus we did not incorporate the known effect that would bias the swirl rotating direction.

8. Swarming and concentrated planktonic *Bacillus subtilis* 3610 show the same motion pattern across different confinement sizes. However, the authors did not give definitive conclusions and evidence. As shown in Figure S1, *Bacillus subtilis* 3610 show completely different cluster behavior. Therefore, the discussion of 3601WT may cause readers' confusion on the article. It may be better to put it in the supporting material.

Another valuable suggestion! We agree that the discussion of *B. subtilis* in contrast with other species of bacteria in the main manuscript may be distracting and confusing. Now, we have moved the part of reporting similar findings of other bacteria and pertinent discussion into the Appendix 1. Presenting data exclusively on SM3 in the main manuscript indeed makes the paper more focused.

9. A central finding of the present study is that the number of vortices/swirls as a function of the well diameter differs for swarming vs. swimming bacteria. The authors argue and show experimentally (Figure 2) that the behavior is identical for small and large diameters. For intermediate values, however, they report that a single swirl is observed for swarming bacteria whereas swimming bacteria show multiple swirls.The fact that the behavior is identical for large wells suggests that the bulk behavior is identical. This is also confirmed by Figure 2E which shows that the spatial correlation function of the velocity is identical in large wells. That suggests that the boundary conditions play a central role in understanding how the observed phenomenology emerges. [Indeed, it was shown in the past that the interaction of bacteria with boundaries crucially determines the formation of swirls in confinement (Lushi, Wieland and Goldstein PNAS 111 9733 2014). The authors of this work assume reflecting boundary conditions, which – to my knowledge – contradicts the finding of Lushi et al.].The authors, however, explain the difference of the observed patterns within their modeling study in a different way, namely by a different strength of the (anti-)alignment interactions. Changing the interaction at the level of individual cells will, however, change the bulk behavior too. Accordingly, the numerically observed bulk behavior (Figure 5B ) is very different in both cases (at a qualitative level). It is difficult to judge the difference in detail because the correlation function was not calculated for the simulations.In short:– The model (Figure 5A) reproduces the experimental results partially (Figure 2C), but the modeling analogue to Figure 2E is missing;– The line of arguments seems not to be entirely consistent.

We agree with the reviewers that sufficiently quantitative comparison between simulation and experiment was lacking in our initial submission. Toward this end, we identified the correlation lengths L_Cv_ from the experimental velocity auto-correlation plots (Figure 2D-E), defined as the point where the velocity correlation (Cv) curve crosses the zero line. It turns out that the correlation length of Cv for swarmer cells is a few microns longer than in planktonic case. This discrepancy persists in all confinement sizes, even at the largest domain size of 500 μm, where L_Cv_ of swarmer cells is ~ 5μm longer than planktonic cells, i.e., ~ 28 μm vs. ~ 23 μm. Guided by this insight and informed by the observation of polar clusters of swarmer cells in dilution experiments, we expanded the simulation work using a new set of simulation parameters (Appendix table 1). We differentiate swarmer cells from planktonic ones by assigning a larger alignment interaction range r_a_ (as defined in Appendix figure 2). The rationale is that stronger interaction between swarmer cells manifests itself as enhanced alignment interaction. The underlying origin for the enhancement of alignment interaction in swarmer cells can be sought in, for example, micro-rheology experiments of the extracellular polymeric network surrounding the swarming bacteria. Such experimental work may be challenging. Nonetheless, the simulation results based on this new set of parameters not only capture the transitional behavior in the experimental VOP diagram, but also provide quantitative agreement between velocity auto-correlation of simulations (new Figure 5B-C) and experiments (Figure 2D-E). Furthermore, it successfully predicts dynamic polar clusters of swarmer cells in dilution experiments (new Figure 5D-E).

We agree with the reviewers that the type of boundary condition determines the formation of swirls in confined systems. Through using a different set of parameters while keeping reflective boundary condition, we found that the present simulation model can reproduce counter-rotating vortices under circular confinement. Therefore, reflective boundary condition per se does not disrupt the formation of counter-rotating swirls in our simulations. However, our extensive PIV analysis does not reveal counter-rotating swirls as those reported in Lushi, Wioland and Goldstein PNAS 111 9733 (2014). Therefore, while we have acknowledged the experimental and simulation results of Lushi et al., we use reflective boundary condition and have focused in this report on the aspect of bacterial collective motion that is caused by differences in alignment interaction range between swarming and planktonic SM3 cells.

10. Inferring the interactions of active particles from observations of the emergent patterns is a highly non-trivial task. In view of this the reviewers are not entirely convinced by the arguments put forward by the authors that "more substantial cell-cell cohesive interaction(s)" are the reason why the swirling patterns formed by swarming/swimming bacteria differ.In this context, we call attention to Ref. [Peruani, Deutsch and Bär: Phys. Rev. E 74 030904(R) 2006]. In this work, a clustering transition of self-propelled rods was described. "Rafts", referred to as clusters by Peruani et al., are observed as the aspect ratio of rods is increased. Notably, a kinetic transition towards clustering can emerge even in the absence of any attractive interactions.In short, the observation that cells move in parallel (polar clusters) next to each other does not allow to conclude that cohesive interactions are present.The Videos S3 and S4 provided by the authors show that the particle shape of swarming and swimming particles is clearly different. In particular, the elongated swarming bacteria show pronounced clusters (Video S3) whereas the shorter planktonic cells (Video S4) do not. The difference in aspect ratio does indeed suggest that swarming and swimming bacteria differ in their alignment interaction. However, this contradicts the observation that spatial correlations in large wells are indistinguishable (see comment 1 and Figure 2E).

The criticism here is profound and insightful. After reading the suggested ref and thinking over the reviewer suggestion, we agree that the formation of the “rafts” does not allow us to conclude that cohesive interactions are present. Thus, we revised the manuscript where we made cohesive interaction statements before and only refer to the interaction as alignment among the swarmers now. In the videos, we agree that planktonic cells appear shorter, and we actually measured their length in Figure 3A. However, we found that CEP treated planktonic cells (Figure 3B-C) and rinsed swarmer cells both with the same average cell length as natural swarmers did not form dynamic clusters at comparable cell density and could not form large single swirls like natural swarmers. Based on these results, we exclude the cell length to be the dominant factor in strengthening the alignment among the swarmers. We also acknowledge that, as a secondary effect, the cell length change may influence the size of the confinement that planktonic SM3 can hold a single swirl within.

On the other hand, we think one possible reason for the alignment decrease of the washed swarmer cells is that after washing, polymers are diluted and the viscosity decreases dramatically in the culture environment of the swarmers, then the momentum of the cells could not be passed as efficiently as previously, resulting in the loss of dynamic clusters and single swirl motion pattern. A recent publication shows that changing the viscoelasticity (by adding purified genomic DNA) of dense swimming *E. coli* alone can turn multi-swirl motion pattern into single swirl motion pattern (Liu, Shankar, Marchetti, and Wu, 2021). We have added reference to this latest report, as it provides a dramatic case consistent with our findings and our own thoughts on matrix viscoelasticity affecting swarm patterns.

Indeed, the spatial correlation in large wells is not totally indistinguishable but shows certain distinction (for swarmers r = 28 um, planktonic cells r = 23 um) and we revised the numerical simulation showing that the difference in alignment interaction range of this magnitude indeed leads to the global motion pattern differences in confinement.

Side remark: in the main text, the authors argue that changes of the aspect ratio are not the reason for an increased alignment interaction, however, in the Discussion section cell morphology changes (e.g. cell elongation and hyper-flagellation) are mentioned as an indicator that swarming is a different phenotype from swimming.

Cell morphology is another aspect of comparison between swarming and swimming phenotypes. The correlation may not necessarily hold, however, between increased alignment and the increase in cell length. Also, these two features are not necessarily to happen simultaneously.